

# Ocean carbon inventory under warmer climate - the case of the LIG

Augustin Kessler[1], Eirik Vinje Galaasen[2], Ulysses Silas Ninnemann[2], and Jerry Tjiputra[1]

[1]Uni Research Climate, Bjerknes Centre for Climate Research, Bergen, Norway
[2]Department of Earth Science, University of Bergen and Bjerknes Centre for Climate Research, Bergen, Norway

*Correspondence to:* Augustin Kessler (augustin.kessler@norceresearch.no)

**Abstract.** During the Last Interglacial period (LIG), the transition from 125ka to 115ka provides a case study for assessing the response of the carbon system to different levels of high-latitude warmth. Elucidating the mechanisms responsible for interglacial changes in the ocean carbon inventory provides constraints on natural carbon sources and sinks and their climate sensitivity which are essential for assessing potential future changes. However, the mechanisms leading to modifications of the ocean's carbon budget during this period remain poorly documented and not well understood. Using a state-of-the-art Earth System model, we analyze the changes in oceanic carbon dynamics by comparing two quasi equilibrium states: the early, warm Eemian (125ka) versus the cooler, late Eemian (115ka). We find a considerable weaker ocean dissolved inorganic carbon (DIC; -314.1 Pg C) storage under the warm climate state in 125ka as compared to 115ka, mainly attributed to changes in the biological pump and ocean DIC disequilibrium components. Due to its large size, the Pacific accounts for the largest DIC-loss, approximately 57% of the global decrease. However, the largest simulated DIC differences per unit-volume are found in the southern sourced waters of the Atlantic. Our study shows that the deep water geometry and ventilation in the South Atlantic is altered between the two climate states where warmer climatic conditions cause southern sourced waters to retreat southward and northern sourced waters to extend further south. This process is mainly responsible for the simulated DIC reduction by restricting the extend of DIC rich southern sourced water reducing the storage of biological remineralized carbon at depth.

## 1 Introduction

The Last interglacial (LIG, or Eemian) is composed of a warm onset around 125ka before present (BP) characterized by warmer temperature in the high latitudes relative to the present and a progressive cooling toward 115ka when the last glaciation was initiated (Otto-Bliesner et al., 2006; Masson-Delmotte et al., 2010). Evidence from land, ice, and ocean records identify the former as the period with the most intense global warming during the last 200,000 years (Turney and Jones, 2010; Dorthedahl-Jensen et al., 2013; Capron et al., 2014) mainly due by changes in the orbital configurations. If the anthropogenic greenhouse gas emissions continue unabated, a climatically anomalously warm state is expected to occur in the near future with a warming that may be equivalent to the high-latitude reconstructed temperature for the LIG (Otto-Bliesner et al., 2013) by the end of this century. For this, the changes in the the warm Eemian period may be considered as an analog for a future warmer climate.

A few studies have examined the carbon cycle dynamics from the LIG period with a particular focus on (1) the ability of models to simulate the transient changes in atmospheric $CO_2$ concentration, which remained relatively stable around 270-280 ppm without displaying any trends and (2) the land carbon budget (Lourantou et al., 2010; Schneider et al., 2013; Brovkin et



al., 2016). Schurgers et al. (2006) used a General Circulation Model (GCM) coupled with a dynamic global vegetation model (DGVM) and a marine biogeochemistry model (HAMOCC3) to analyze the global carbon dynamics. Their simulated trend in atmospheric $CO_2$ concentration diverges from that recorded in the ice core data and shows a constant increase for the 128-113ka period. This was mainly attributed to the simulated decrease in the terrestrial carbon storage of 350 PgC. Brovkin et al.

(2016) used three different Earth system Model of Intermediate Complexities (EMIC; Bern3D-LPJ, CLIMBA and GENIE). The computationally efficient EMIC models explicitly simulate interactions between all Earth system components but in a more parametrized form, allowing for long-term transient simulations. However, they have limitations in assessing climate change processes at regional scale. Thereby Brovkin et al. (2016) analyzed and compared the carbon cycle dynamics during the Holocene and the Eemian by applying the same set of forcings to these two periods. They could qualitatively explain carbon

dynamics during the Holocene but as with Schurgers et al. (2006), their simulated atmospheric $CO_2$ diverges after 121ka from that of the data. They suggested that the forcings ($CaCO_3$ shallow water accumulation and natural terrestrial carbon changes) were rather unrealistic to apply for the Eemian period and the absence of permafrost module could lead to an inability of the model to properly respond to cooling. Kleinen et al. (2016) simulated atmospheric $CO_2$ within the range of the ice core data as compiled by Bereiter et al. (2015) using the EMIC CLIMBER2-LPJ, which includes shallow water coral $CaCO_3$ sedimentation

and peatland dynamics. In their experiment, the ocean acts as a source of $CO_2$ to the atmosphere, with outgassing rates that range from $0.35 \, PgC \, yr^{-1}$ in 126ka to $0.17 \, PgC \, yr^{-1}$ in 116ka. This outgassing is mainly attributed to the strong $CaCO_3$ coral formations during the first half of the period, induced by sea level rise, which counterbalances the weathering and peatland fluxes.

While the above studies provide a better understanding of land carbon budgets, the mechanisms leading to changes of

the ocean's carbon budget, and more generally marine carbon and nutrient cycling, during the Eemian period remain poorly documented and not well understood. Elucidating the mechanisms responsible for changes in the ocean carbon distribution and inventory is of interest as it provides past constraints and context for evaluating the response of natural carbon sources and sinks to future climate change. This study aims to fill this knowledge gap by analyzing and comparing, in terms of ocean carbon dynamic, two opposite states of the LIG: the early and warm Eemian onset (125ka) versus the cooler and late Eemian

(115ka). Using a state-of-the-art Earth System model, our study addresses the regional differences in the ocean carbon storage and the underlying mechanisms.

The paper is organized as follows: in Section 2, we describe the model, the experiment design, as well as the terms and metrics used to quantify the differences in carbon dynamics during the two periods. Section 3 presents the results of the model simulations, while discussions and comparison with previous studies are presented in Section 4. Finally, the study is

summarized in Section 5.



## 2 Method

### 2.1 Model description

The present study uses output of an updated version of the Norwegian Earth System model (NorESM1-ME), which has been recently developed to efficiently perform multi-millenial and ensemble simulations (Bentsen et al., 2013; Luo et al., 2018). This model includes an isopycnal-coordinate ocean general circulation model based on the Miami Isopycnic Coordinate Ocean Model (MICOM, Bleck et al., 1992) and a biogeochemical ocean module adapted from the Hamburg Oceanic Carbon Cycle (HAMOCC5) model (Maier-Reimer, 1993; Maier-Reimer et al., 2005; Tjiputra et al., 2013). The inorganic seawater carbon chemistry in HAMOCC5 includes prognostic partial pressure of $CO_2$ (p$CO_2$) according to the Ocean Carbon-Cycle Model Intercomparison Project (OCMIP) protocols. The p$CO_2$ is computed as a function of temperature, salinity, dissolved inorganic carbon (DIC), total alkalinity (TALK) and pressure. This adapted version of HAMOCC5 does not include prognostic weathering fluxes, but employs a 12-layers sediment model following Heinze et al. (1999), which is particularly relevant for long-term transient simulations. The horizontal resolution of the land and atmospheric components is approximately $2°$, while the ocean and ice components have higher resolutions of approximately $1°$. In the vertical, the ocean model adopts 53 isopycnal layers.

The land component in NorESM (CLM4, Community Land Model version 4) is based on version 4 of the CLM family (Lawrence et al., 2012a). The land surface is sub-gridded into three sub-gridded entities: land units, columns and plant functional types (PFTs). These sub-gridded cells are used to represent large-scale patterns of the landscape, variability in the soil and snow state variables, and the exchanges between land surface and atmosphere, respectively. Each of the sub-grid entities has its own prognostic variables, is independent and experiences the same atmospheric forcing. Each cell is averaged and weighted with its fractional area.

The marine ecosystem is based on a Nutrient-Phytoplankton-Zooplankton-Detritus (NPZD) model that includes dissolved organic carbon (DOC). The inorganic nutrients consist of three macronutrients (phosphate, nitrate and silicate) and one micronutrient (dissolved iron). A constant Redfield ratio is adopted in the model as $P : C : N : \triangle O_2 = 1 : 122 : 16 : -172$. The phytoplankton growth rate is expressed as function of temperature, light (Smith, 1936; Eppley, 1972), phosphate, nitrate and dissolved iron availability, and its loss is regulated by an exudation and mortality rate, in addition to zooplankton grazing. The penetration of light decreases with depth following an exponential function, which responds to a gradual extinction factor formulated as a function of water depth and chlorophyll concentration (Maier-Reimer et al., 2005). The model prescribed a global constant vertical sinking speed of particles produced in the euphotic zone (>100 m depth). The particulate organic carbon (POC), which comprises dead phytoplankton and zooplankton, sinks through the water column with a speed of 5 m day$^{-1}$ and is remineralized at a constant rate of 0.02 day$^{-1}$ when oxygen is available. Other particles such as opal shells and particulate inorganic carbon (PIC) sink at a speed of 60 and 30 m day$^{-1}$, respectively. Particulates that reach the sea floor without being remineralized interact chemically with the sediment pore water via bioturbation and vertical advection. In the model, the air-sea gas exchange of $CO_2$ and $O_2$ only occurs between the ocean surface and the atmosphere in ice-free regions and is computed according to the following three components (Wanninkhof, 1992): the gas solubility in seawater, which is computed as a function of surface salinity and temperature according to Weiss (1970, 1974); the gas transfer velocity, which



is proportional to the square of the surface wind speed and is computed as a function of the Schmidt number; and finally the air-sea gradient of gas partial pressures. To better elucidate various biogeochemical processes on the carbon cycle, the model is updated to also include preformed $O_2$, TALK and $PO_4$ tracers in the biogeochemical module. Finally, in order to provide information of the water mass ages since its last contact with the atmopshere, an idealized age tracer is implemented and sim-

ulated in the NorESM model. Hence, the age tracer is set to zero for all water masses at the ocean surface and subsequently transported and mixed passively with circulation in the ocean interior and integrated with the model time step. This tracer is also used to estimate the the ventilation rate of different interior water masses.

## 2.2   Experiment setup

Two equilibrium experiments are performed over the Eemian, one near the onset (warmer than today; 125ka) and one at the end

(colder; 115ka) of the Last Interglacial. Both experimental configurations follow the standard protocols of the third phase of the Paleoclimate Modelling Intercomparison Project (PMIP3; $URL = https : //pmip3.lsce.ipsl.fr/$), with a fixed vegetation coverage from the pre-industrial boundary conditions. The only differences with the pre-industrial configurations are the orbital parameters and the greenhouse gases concentrations ($CO_2$, $CH_4$, $N_2O$). For the experiment at 125ka (115ka), the atmospheric $CO_2$, $CH_4$ and $N_2O$ levels are prescribed to be 276 ppmv (273 ppmv), 640 ppb (472 ppb) and 263 ppb (251 ppb), respectively.

The two experiments are branched off from 1000 years of spin up with a pre-industrial set up and forced with their respective interglacial boundary conditions for 4000 simulation years.

In the last 50 years of Eemian forcing simulations the ocean is close to equilibrium. Only small drifts remain, mainly in the Pacific basin where the equilibrium is still not fully established. Therefore, the global ocean DIC and TALK slightly decrease in 125ka (115ka) experiment by approximately $-0.15$ Pg C yr$^{-1}$ ($-0.06$ Pg C yr$^{-1}$) and $-0.01$ Pmol yr$^{-1}$ ($-0.01$ Pmol yr$^{-1}$),

respectively. However, these drifts are small compared to the absolute ocean budget in DIC (37391 and 37705 Pg C) and TALK (3291 and 3303 Pmol) for the experiment 125ka and 115ka, respectively. The $CO_2$ flux is relatively constant and depicts the ocean as a weak source to the atmosphere with an outgassing of $0.12 \pm 0.06$ in 115ka and $0.15 \pm 0.06$ Pg C.yr$^{-1}$ in 125ka.

## 2.3   DIC decomposition

In order to analyze the oceanic carbon cycle, the dissolved inorganic carbon (DIC) is decomposed into three DIC components

(Eq. (1)), of saturated, biological and disequilibrium components following Bernadello et al. (2014):

$$DIC^{tot} = DIC^{sat} + DIC^{bio} + DIC^{dis} \tag{1}$$

The DIC at saturation (DIC$^{sat}$) describes the DIC concentration when the water parcel is in full equilibrium with the atmospheric $CO_2$ when it is last in contact at surface. This variable is computed offline with the inorganic carbon chemistry program CO2SYS developed in Matlab (van Heuven et al., 2011) using the model output of preformed alkalinity ($TALK^{pre}$),

preformed phosphate ($PO_4^{pre}$), surface silicate, salinity and temperature. In addition, the atmospheric $CO_2$ concentration from



each experiment is used. To complete the CO2SYS input, we applied the dissociation constants K1 and K2 introduced by Mehrbach et al. (1973) and refitted by Dickson and Millero (1987).

The biological component of DIC comprises (1) the interior remineralization of organic matter (expressed in carbon), which is produced in the euphotic layer via photosynthesis (also referred to as soft-tissue pump) and (2) the remineralization of

planktonic calcium carbonate shells (expressed in carbon; calcium carbonate pump). These two remineralization components are added to form the biological component of DIC, as shown in Eq. (2).

$$DIC^{bio} = DIC^{soft} + DIC^{carb}. \tag{2}$$

The remineralization of soft tissues (hereafter $DIC^{soft}$) contributes via phosphate ($PO_4$)-remineralization through a carbon phosphorus stoichiometric ratio $r_{C:P} = 122$. This component is calculated from the difference between the total and the

preformed $PO_4$ according to

$$DIC^{soft} = r_{C:P}(PO_4^{tot} - PO_4^{pre}). \tag{3}$$

The carbonate pump contributes through the dissolution of $CaCO_3$ hard shells, calculated as difference between the total and the preformed alkalinity and $PO_4$ following

$$DIC^{carb} = 0.5[TALK^{tot} - TALK^{pre} + r_{N:P}(PO_4^{tot} - PO_4^{pre})], \tag{4}$$

where $r_{N:P} = 16$ is the Redfield ratio adopted by the model and the phosphate term accounts for the alkalinity changes owing to the soft-tissue pump.

Finally, the disequilibrium component of DIC ($DIC^{dis}$) measures the disequilibrium state of the surface water with respect to the atmosphere. This parameter is computed from the difference between the $DIC^{tot}$ (output) and the other DIC components previously mentioned in Eq. (1). A negative $DIC^{dis}$ occurs when the water parcel sinks into the ocean interior before a full

equilibration with the atmosphere is obtained, which lead to an undersaturation of the water parcel. This undersaturation can also be reinforced by biological $CO_2$ consumption at the surface, which tends to increase the time scale needed for the water parcel to equilibrate. On the contrary, a positive $DIC^{dis}$ translates into a supersaturation. This latter occurs when deep waters, which contains high concentration of DIC because of remineralization processes, upwell or mix vertically with the surface waters (Follows and Williams, 2004). Both, $DIC^{dis}$ and $DIC^{sat}$ are transported by ocean circulation into the interior ocean.

In our analysis, we mostly show differences between the warmer 125ka and the colder 115ka experiments. We therefore use the delta notations $\triangle DIC^{tot}$, $\triangle DIC^{sat}$, $\triangle DIC^{soft}$, $\triangle DIC^{carb}$ and $\triangle DIC^{dis}$ to refer changes between the warmer and the colder periods.





## 2.4 Water mass analysis

In order to identify water mass sources, we apply the 'PO' tracer as defined by Broecker (1974). It is computed using phosphate and oxygen fields following

$$PO = O_2 + r_{O:P} \times PO_4 \tag{5}$$

where $r_{O:P} = 172$ is the phosphorus to oxygen stoichiometric ratio used in the model. This tracer is presumed to be nearly constant for a specific water mass. It is based on the principle that phosphate is released, while oxygen is used during remineralization, and vice versa during biological production. The distinction of water masses using PO is useful for contrasting water masses with very different surface PO values. Here, we mainly use PO to identify northern- and southern-sourced water masses (NSW and SSW) in the deep ocean below 1000 m depth characterized by low and high PO values, respectively.

## 10   3   Results

We will first describe near surface changes that particularly influence the biological pump. The second section addresses the differences in water mass properties. Finally, we describe and summarize the overall changes in the global and regional oceanic DIC storage. Each analysis have been performed over the average of the last 50 years of the simulations. In addition, we divided the global ocean into three main basins (Atlantic, Indian and Pacific).

## 15   3.1   Near surface productivity

Simulated sea surface temperature (SST) during the 125ka experiment were warmer globally, but the changes varied spatially and seasonaly, affecting both the ventilation and the nutrient supply at the surface. Warmer (cooler) SSTs lead to a more (less) stratified and lower (higher) nutrient concentration in the surface ocean, due to a weakening (strengthening) of the mixing process. In the Atlantic, cooler SSTs are simulated during boreal winter and spring ($\triangle SST < 0$, Fig. 1a-b), which allow
for more upwelled nutrients to the surface (Fig. 2a) via an increasing of the mixed layer depth. This higher concentration of nutrients increases the biological production (Fig. 2b) under more favorable warmer blooming season in summer in 125ka ($\triangle SST > 0$, Fig. 1c).

    In the Atlantic section of the Southern Ocean (SO), colder SSTs throughout the year are simulated during 125ka in some sections of the subantarctic ($45°S$ latitude band) corresponding to southern sourced intermediate water formation region in
the model (Fig. 1). This colder water is associated to stronger winter mixing, which leads to water mass with high preformed nutrients (Fig. 2a, green rectangle). This water mass sink and reemerges along the Equator leading to an increased biological and hence export production in this region (Fig. 2b, green rectangle). A similar 'ocean tunnel' (Fig. 2a-b, purple rectangle) connects the high and low latitude Pacific but results in the opposite sign of change. Here the SO is simulated as having warmer SSTs throughout the entire year (Fig. 1a-d). This leads to a lower preformed nutrient concentration in 125ka (Fig. 2a, purple
rectangle in the Pacific SO) negatively affecting the biological productivity in the equatorial upwelling regions where these



waters return to the surface (Fig. 2b, purple rectangle at the Equatorial Pacific). Thus, the simulations reveal that changes in southern hemisphere thermocline ventilation regions modulate basin scale productivity and export production even within an interglacial period with modest changes in external forcing. This result is broadly consistent with previous studies suggesting that the upper limb of the biogeochemical divide is critical for setting biological export production and is sensitive to climate

changes (Sarmiento et al., 2004; Marinov et al., 2006; Moore et al., 2018). Despite latitudinally homogeneous forcing we find a basinally heterogeneous response in both subantarctic ventilation and in low latitude productivity which is similar to, albeit more extreme than, the basin specific response simulated for future warming and stratification (Moore et al., 2018).

There are no significant changes in the biological activity in the Indian Ocean. However, a weak decrease in phosphate availability is simulated (Fig. 2a), probably induced by warmer SSTs during the boreal fall season leading to less nutrient

upwelling to the surface. Here the carbon export is slightly weaker in 125ka and more particularly around $40°$ S and the Arabian sea where the waters upwell.

In addition to variation in ventilation rates, these changes in surface physical and biological activities could also have implications on the exchanges of carbon between near-surface and interior water masses, and therefore the interior carbon budget. In the next section, ventilation changes in 125ka are compared to 115ka by analyzing the simulated water mass properties.

## 3.2   Water mass properties

The analysis of the water mass age allows us to examine the interior ocean ventilation rate. A reduction in water mass age translates to a stronger ventilation rate age and vice versa for an increase in age. The differences in water mass age between 125ka and 115ka are presented in Fig. 3 depicting the zonally averaged sections for each ocean basin. The water mass ages in the Atlantic and the Indian show similar patterns with mean older water masses in the upper layers at 125ka (roughly

$+100$ years), and younger water masses below 1000 m depth (by as much as $500$ years younger). The Southern Ocean (south of $50°S$) contains younger water masses throughout the entire water column in both basins at 125ka, suggesting a stronger ventilation rate. However, this does not stem from temperature changes since the SST in the SO is rather warmer in 125ka (Fig 1). Instead, this is likely due to changes in the SSW to NSW distribution. Figure 4 shows that there is a clear distinction between interior water mass structure in the Atlantic between 125ka and 115ka. It shows that the SSW retreats further southward in

125ka relative to 115ka in the Atlantic. This confinement in SSW is induced by the change in the Antarctic sea-ice cover. Such processes have been introduced byFerrari et al. (2014). In addition, using the same model simulations as the present study,Luo et al. (2018) show in the supplementary information Fig. S8 that the surface wind speed in the east and west southern Atlantic are relatively similar in 125ka and 115ka, translating a relative unchanged Antarctic circumpolar current between the two periods which can therefore not explain the SSW retreat. However, the model also simulates a modification in SSW density ($-0.2$ kg

m$^{-3}$ in 125ka compare to 115ka, Fig 4). This reduction in water density is mainly driven by the input of low salinity fresh water from the melting of the Antarctic sea-ice and may have an additional impact on determining the Atlantic distributions of NSW and SSW. As a net result, the water mass becomes younger in the SO because of a southward retraction of SSW and southward incursion of more and younger NSW.



While such large redistributions of northern and southern origin deep waters only occurs in the Atlantic, these changes also influence water properties in the Indian Ocean due to the 'downstream' advection of younger deep water into the interior during the warmest period (Fig 3b). In addition to simple advection of younger water northward in the Indian, the residence time (turnover rate) of Indian deep water must also decrease (increase) since the ventilation ages decrease northward at depth.

By contrast, in the Pacific, the zonally averaged bottom water mass ages are simulated to be older in 125ka (Fig 3c). However, this basin can be divided between the western and eastern side. While the western side is also influenced by the younger water masses created in the Atlantic, the eastern side waters of the basin are simulated to be older in 125ka by as much as 300 years older. This older water masses are created in the Pacific SO and are predominantly affected by the strong increase in SST, increasing therefore the stratification. In the northern hemisphere the younger waters are due to to cooler SST (Fig. 1).

The southern sourced waters are particularly affected in terms of geometry distribution. We therefore divided the changes from these SSW into the three basins. Table 1 summarizes the changes occurring below 1000 m depth in the SSW in terms of volume, DIC and water mass age for each basin and reveals the Atlantic as the most affected area under warmer climate conditions. The relative difference in all those three characteristics ($\triangle V_{SSW}$, $\triangle Age_{SSW}$ and $\triangle DIC_{SSW}$) between 125ka and 115ka are the greatest in the Atlantic ($-37\%$ , $-262$ years and $-0.92$ g C m$^{-3}$). This demonstrates that the ventilation

mechanism in the Atlantic sector of the SO is likely to be more sensitive (than in other basins) to climate change.

They are significantly different and basin specific responses of deep water ventilation rates and water mass distribution to uniform changes in forcing. In the next section we discuss the global and regional carbon budget differences between 125ka and 115ka periods associated to these responses.

## 4   Global and regional carbon budgets

Figure 5 shows the difference in the carbon inventory vertical profiles between 125ka and 115ka as simulated by our model. The changes in DIC$^{tot}$, DIC$^{sat}$, DIC$^{soft}$, DIC$^{carb}$ and DIC$^{dis}$ are averaged over a 500 m depth interval. The global amount of DIC$^{tot}$ is $-314.1$ PgC in the ocean under the warmer condition (Fig. 5a, gray $\triangle DIC^{tot}$). Here, the Atlantic accounts for 15% ($-49.1$ PgC) of that global decrease, while the Indian and the Pacific basins contribute to 28% ($-87.2$ PgC) and 57% ($-179.0$ PgC), respectively. Only in the near-surface layers the model simulates a positive $\triangle DIC^{tot}$, which translates to

higher surface DIC concentration at 125ka relative to 115ka. Most of the ocean interior has lower DIC concentration at 125ka with the strongest difference in $\triangle DIC^{tot}$ simulated between $2000-3000$ m depths for each basin (Fig. 5b-d). The soft-tissue pump and the disequilibrium effect are the main contributors for the weaker carbon inventory depicted at 125ka at global scale (Fig. 5a, green and purple bars) - each accounting for a third of the total $-314.1$ PgC decrease. Similarly, $\triangle DIC^{tot}$ in the Atlantic basin is also predominantly controlled by the biological pump, i.e. the soft-tissue plus carbonate pump throughout the

entire water column with a decrease at depth and an increase in the near-surface (Fig. 5b). Except for the upper ocean, the contribution from saturation component related to temperature and salinity change is generally negligible.

The $\triangle DIC^{tot}$ of the Indian basin resembles that of the Atlantic at depth, where the soft-tissue and disequilibrium components simulate the strongest decrease (Fig. 5c, green and purple bars). However, the saturation component depicts persistent



negative $\triangle DIC$ below 1000 m depth, thereby accounting for the second most important component of the decrease throughout the water column. In addition, near-surface changes in $\triangle DIC^{tot}$ are controlled by the changes in $\triangle DIC^{sat}$. Similarly, the near-surface layer of the Pacific is also controlled by the changes in saturation component (Fig. 5d), simulating a strong positive difference of about $+18$ PgC. Changes in the deeper layers are mainly attributed to the disequilibrium effect and the soft-tissue

pump accounting for a decrease of $-83.6$ PgC and $-44.0$ PgC in 125ka relative to 115ka, respectively. However, the saturation component has also a considerable influence on the carbon storage with persistent negative $\triangle DIC^{sat}$ throughout the water column.

In order to address the regional changes, we analyze the differences in each DIC components further by calculating the zonally averaged values in each basin. Figures 6, 7 and 8 depict these differences for the Atlantic, Indian and Pacific basins,

respectively. As shown in Fig. 5a, the carbon inventory of the Atlantic is reduced mainly below 1500 m depth. Here the southern hemisphere is the most affected region, which depicts the strongest differences in $\triangle DIC^{tot}$ (Fig. 6a, blue shades). This pattern corresponds well to the changes in soft-tissue pump (Fig. 6b). Near the surface, the higher carbon export mentioned in Sect. 3.1. increases the remineralization of organic matter leading to higher DIC concentration in 125ka. At depth, the changes in SSW and NSW lead to a decrease in younger water masses, hence less remineralized organic matter. This is reflected by the

negative $\triangle DIC^{soft}$ and $\triangle DIC^{carb}$, translating to a less effective soft-tissue and carbonate pump in 125ka. Positive change in $\triangle DIC^{tot}$ also arises from the soft-tissue and carbonate signal due to the increase of the alkalinity (not shown here) and slightly older water masses along the African coast. The bottom waters in the southern hemisphere are mainly controlled by a stronger disequilibrium effect, i.e. negative change in comparison to 115ka. The latter is due to the retreat of the SSW and the inflow of more NSW between $50°S$ and the Equator in 125ka. The NSW water mass, formed in the North Atlantic, is generally

more subject to biological production during its near surface northward transport before sinking into the interior than the SSW. The biological production consumes DIC during photosynthesis and pushes the water mass further out of the equilibrium with the atmospheric $CO_2$, inducing $\triangle DIC^{dis}$ to be more negative. The negative values of $\triangle DIC^{dis}$ are conserved when the water parcel flows southward into the deep ocean. For this reason, the regions that are no longer influenced by SSW in 125ka depict a negative $\triangle DIC^{dis}$ (Fig. 6d) . However, the upper layers of the north Atlantic are simulated with higher $DIC^{dis}$ (positive

$\triangle DIC^{dis}$) due to weaker ventilation induced by stronger SSTs in the Labrador and Nordic sea. Finally, the loss of carbon in the Southern Ocean is shown to be mainly attributed to a decrease of the saturation component in 125ka. This decrease is attributed to lower salinity and TALK$^{pre}$ (not shown here) provoked by the melting of the sea-ice.

The DIC storage in the Indian generally shows a decrease in 125ka with the strongest changes occurring at depth north of $30°S$ (Fig. 7a, dark blue shade). Only in the region that may correspond to the AAIW the simulated $\triangle DIC^{tot}$ are positive

(Fig. 7a, red shade). Similar to the Atlantic, the pattern of the soft-tissue pump changes corresponds to the $\triangle DIC^{tot}$ pattern throughout most of the Indian basin. This decrease in biological remineralization is in agreement with the water mass age changes seen in Sect. 3.2 (Fig. 3b): younger water masses account for less biologically-induced DIC content. However, the bottom and the surface waters show opposite signs in the $\triangle DIC^{tot}$, which suggests that other processes are acting in these regions. The differences in the carbonate pump remain small and roughly follow the pattern of the soft-tissue pump (Fig. 7c).

Changes in the bottom water $DIC^{soft}$ can be attributed primarily related to the difference in the disequilibrium effect due





to stronger carbon export in the Southern Ocean (Fig. 2b) and to a slight decrease in the saturation component. The strong positive $\triangle DIC^{dis}$ simulated at near-surface along the Indian coast is well in agreement with the strong reduction in primary productivity. However, this is not shown in the soft-tissue pump because of the cooler SSTs described previously, increasing the ventilation and the DIC supply from DIC-rich deep waters. Finally the negative $\triangle DIC^{sat}$ depicted in the top layers in the

north of the Indian Ocean is mainly attributed to a change in water mass origin from 115ka to 125ka. During 115ka the SSW upwells from the deep ocean into the Arabian sea. By contrast, at 125ka, the waters coming from the Indonesian region mix with SSW. These Indonesian throughflow water masses initially coming from the Pacific are affected by strong precipitation in the Indonesian basin, which reduces the ALK.

The Pacific shows the strongest DIC$^{tot}$ decrease in the northern hemisphere, mainly due to the reduction in soft-tissue

pump (Fig. 8b). This lower organic remineralization arises from an increase of the ventilation around $30°N$ and a potential increase of the upwelling generating younger water masses (Fig. 3c). This is in good agreement with the increased carbon export production (Fig. 2b), inducing positive $\triangle DIC^{tot}$ near the surface, and with the cooler SST depicted in Fig. 1. The DIC inventory of the southern hemisphere bottom and near-surface waters is larger in 125ka relative to 115ka. This is also mainly due to the changes in soft-tissue pump, which is more effective in 125ka both due to longer residence time of the water

masses (Fig. 3c) and increased biological export production during the Austral spring. The older water masses as mentioned in Sect. 3.2 also suggest that the ventilation of the Eastern bottom water in the South Pacific is slower in 125ka, mainly attributed to the warmer SSTs as mentioned in Sect 3.1. (Fig. 1). The carbonate pump has a relatively low impact on the total $\triangle DIC$ inventory in the Pacific but follows the same pattern than the changes in the organic carbon remineralization. The disequilibrium effect accounts for the strongest decrease of DIC throughout the basin as seen by the negative $\triangle DIC^{dis}$

in almost all regions (Fig. 8d). It can be attributed to the higher biological productivity and export in 125ka in the Southern Ocean. This SSW mainly composes the Pacific interior ocean. Finally, the saturation component is controlling the changes occurring in the near-surface waters (Fig. 8e). Lower saturations are attributed to higher export of calcium carbonate south of $60°S$, which lower the alkalinity at the surface and thereby the buffering capacity. On the other hand, the calcium carbonate formation decreases north of $60°S$, resulting in a higher alkalinity and buffering capacity.

## 5   Discussion

The ocean plays an important role in storing carbon and, thus, in the long term regulation of atmospheric $CO_2$ levels. The processes involved in regulating the ocean carbon inventory are likely to change under warmer future conditions. In this study, we simulate two equilibrium states of the penultimate interglacial period using a state-of-the-art Earth System Model and make a first attempt at quantifying the biogeochemical and physical processes responsible for carbon storage changes

caused by different (interglacial) orbital configurations and background climates. Significant decreases of the ocean carbon storage capacity are found under a warmer climate. Most of this decrease is induced by the reduction of the biological pump. This decrease is found to be mainly driven by the shorter residence time of interior deep water masses. Additionally, spatial





modifications in ventilation structure are shown to be also responsible for this carbon change, by impacting the biological pump and hence the remineralization process.

Using the available proxy reconstructions during the LIG period allows us to assess the validity of important features in our model results. We assess the validity of the simulated 115ka to 125ka water mass geometry change using LIG proxy reconstructions of bottom water $\delta^{13}C$, a water mass tracer strongly but inversely related to carbon and PO$_4$ contents (Eide et al., 2017). Similar to our results, expanded SSW in the late compared to early LIG has been previously been inferred from such reconstructions (Govin et al., 2009). Records of bottom water $\delta^{13}C$ indicate less influence of (high-$\delta^{13}$C) NSW in the deep South Atlantic at 115ka (lower $\delta^{13}C$) than at 125ka (higher $\delta^{13}C$) while mid-depth North Atlantic NSW influence (high $\delta^{13}C$) remained largely unchanged (Fig. 9). This pattern of $\delta^{13}C$ in the deep South Atlantic (Site 1089, 4.6 km water depth) diverging from the mid-depth North Atlantic (Site 983 and JPC8, 2 km water depth) indicates relatively greater SSW influence in the abyssal South Atlantic at 115ka, while $\delta^{13}C$ intermediate between these two indicates an mixture of SSW and NSW shallower in the South Atlantic (Site 1090 and MD07-3077, 3.8 km) (Fig. 9). This 115ka to 125ka water mass geometry change inferred from reconstructions is strikingly similar to our model results, suggesting that in the colder (115ka) climate NSW ventilation persisted while SSW expanded northward in the abyss on millennial timescales (Fig. 4). Also consistent with our results, ice core proxies indicate that Southern Ocean sea ice extent was greater at 115ka than at 125ka (Wolff et al., 2006; Röthlisberger et al., 2008), while our model reproduces the volumetric SSW expansion in response to this increase in Southern Ocean sea ice extent as suggested for glacial climates (e.g. Ferrari et al. (2014)). Our model results (Fig. 4) suggest similar sea ice and SSW expansions, albeit muted compared to glacial changes, occurred in response to LIG orbital configuration changes and without continental ice sheet growth (not included in the model), indicating a relatively tight coupling between Antarctic climate, sea ice, and the deep Atlantic water mass geometry changes influencing ocean carbon storage.

The changes in ocean carbon storage simulated by our model are significant and demonstrates that warm (interglacial) ocean carbon content changes with climate forcing. While atmospheric CO$_2$ is fixed in our model preventing a direct assessment of ocean carbon changes on atmospheric CO$_2$, the decrease in deep carbon storage and shoaling of the DIC pool during the warm 125Ka interval is generally consistent with higher atmospheric CO$_2$ levels at this time. A sense of the scale of the changes our model simulations can be gained through comparison to previous modelling efforts where atmospheric CO$_2$ was not fixed. Brovkin et al. (2016) also simulated a decrease in DIC$^{tot}$ under warmer Eemian conditions using simpler EMIC models, but at slightly weaker decrease than in our study. In their study, the contribution of the ocean to the change in atmospheric CO$_2$ concentration ranged from 11 to 41 ppm between 126ka and 115ka translating a difference of DIC storage capacity of about 22 to 82 PgC – four times smaller than our finding. Their simulated change in atmospheric CO$_2$ after 121ka was in the opposite direction (increasing) than that of the atmospheric trend observed in ice core data. Hence, assuming their vegetation model is correct, this suggests that to keep the atmospheric CO$_2$ level within the observed range, the ocean needed to take up more carbon at the end of the LIG to counteract terrestrial reservoir changes. In other words, the difference in ocean carbon storage they simulated should be larger, in accordance with our study. Thus, both studies suggest that ocean carbon storage must have been the dominant factor driving changes in atmospheric CO2 concentrations during the LIG.





Another study by Schurgers et al. (2006) obtained a difference in atmospheric $CO_2$ concentration and terrestrial carbon storage of about 40 PgC and 350 PgC, respectively, between the onset and end of the LIG. This potentially translates to a 310 PgC of difference in ocean carbon storage, which corresponds well in absolute magnitude to our findings of -314.1 Pg C. However, the changes in $CO_2$ concentration in the atmosphere that they simulated steadily increases, which potentially points

toward more carbon needing to be stored in land or ocean toward the end of the LIG. Finally, these carbon dynamic focused studies demonstrate a weakening in the capacity of the ocean to store carbon in the beginning of the Eemian period under warmer climate conditions, which is supported by our study.

Concerning the modification in the upper ocean productivity under warmer climatic conditions, our model study shows an heterogeneous response in phosphate availability and carbon export production especially between the Atlantic and Pacific

basins. Such heterogeneous response of the biogeochemical divide have also been highlighted by Moore et al. (2018) for future projections under warmer climatic conditions. This implies that changes in the biogeochemical divide could somewhat be similarly impacted from past and future anthropogenic $CO_2$ forcings. Reconstruct and understand the pattern and signs of past responses of large scale productivity to climate forcing are therefore critical for assessing not only the sign but also the sensitivity of different regions to climate change.

There are limitations to our study. Factors that could influence ocean carbon storage including sea level, riverine input of nutrients, and atmospheric dust loading, which are all set to preindustrial levels in our simulations, but may have been different in the LIG. Global sea level, for example, may have been as much as 6-9 m above present (Kopp et al., 2009). Further, we compare two quasi-equilibrated states, which is unrealistic and ignores transient forcings and shorter-term variability. This may explain differences between our model results and some proxy reconstructions. For example, proxy reconstructions suggest

that both NSW and SSW ventilation may have varied considerable near 125ka (Galaasen et al., 2014; Hayes et al., 2014). The changes suggested by these studies include reductions of NSW and expansions of SSW similar to our modeled 115-125ka equilibrium difference, but then occurring as short-lived (centennial-scale) transient events associated with freshwater input episodes during the final phase of northern deglaciation (Galaasen et al., 2014). Our quasi-equilibrated model simulations for 115ka and 125ka, also lack ice sheet and the corresponding freshwater input variability, do not address such shorter-term

changes that could affect the ocean carbon inventory (Stocker and Schmittner, 1997). However, short-lived changes would likely have less impact on the ocean carbon inventory than the longer-term (millennial-scale) changes we address, the latter allowing all carbon system components and ocean dynamics to adjust. Thus, we still expect our model simulations to provide insight into baseline changes and redistribution of ocean DIC forced by the different LIG orbital configurations, supported by the important role of deep Atlantic water mass geometry changes coupled with its similarity to the long-term (millennial-scale)

evolution inferred from proxy reconstructions (Fig. 4; Fig. 9).

## 6   Conclusions

The ongoing global warming raises questions about the oceanic carbon sink and its efficiency under a warmer climate condition. In this study, we use a fully-coupled NorESM model to simulate two quasi-equilibrium states of the Last Interglacial: one period



is globally colder (115ka) and one is globally warmer (125ka) than today. We focus on the differences that occurred in 125ka in comparison to 115ka, specifically the differences at global and basin scales. Thereby, to our knowledge, it is the first attempt in elucidating the biogeochemical and physical processes that are responsible for the ocean carbon inventory changes under warmer climate conditions during the LIG.

We found that the global ocean carbon budget decreases during the warm period (-314.1 PgC). The Pacific Ocean has the largest reduction and accounts for 57% of the global DIC loss, while the Indian and Atlantic basins account for 28% and 15%, respectively. However, these quantities mostly reflect basin volumes. The southern-sourced waters (SSW) are revealed to play an instrumental role for the DIC changes in the interior below 1000 m depth. In these waters, the Atlantic is highlighted to be the region where the strongest DIC loss occur per unit volume and is characterized by a stronger ventilation and a $DIC^{tot}$

decrease of about 37% compared to its respective value in 115ka.

    The reduced DIC budget in 125ka occurs mostly in the interior ocean, while there is a weak increase in the top 1000 to 1500 m depths. Two factors contribute mainly to the drop in the DIC budget in the interior ocean are (1) a weaker biological component from both the soft-tissue and the carbonate pumps that dominates at the depth between 1000 to 3000 m, and (2) a stronger disequilibrium effect (i.e. more negative) of DIC in the bottom waters. The latter is predominantly affected by changes

in biological production at the surface. Stronger biological production during warmer period pushes the surface water out of equilibrium with the atmospheric $CO_2$ and vice versa during the colder period. These modifications in biological productivity are, however, heterogeneous between basins. While the Atlantic accounts for more biological production in 125ka, the Pacific productivity decreases.

    The weakening of the biological component at depth is driven by younger water masses simulated in the interior ocean. This

decrease in residence time of the water masses is provoked by the strong SST modifications that affect the ventilation in 125ka as compared to 115ka. Higher SST, especially in the high latitudes, induces strong summer sea-ice retreat in the Atlantic sector of the Southern Ocean and stratification in the Pacific Ocean. In the Atlantic, this results in a more southerly confined SSW and southward expansion of NSW in the deep ocean. These water masses are advected by the Antarctic circumpolar current into the Indian and the eastern Pacific. The western Pacific is influenced by water masses coming from the Pacific Southern Ocean,

with a warmer SST that hinders the ventilation and increases the residence time of the interior water masses on the eastern side of the basin.

    Concerning the modification in the upper ocean productivity under warmer climatic conditions, our model study reveals clear yet heterogeneous changes in phosphate availability and carbon export production especially between the Atlantic and Pacific basins. Such inter basinal response in the biogeochemical divide has also been highlighted by Moore et al. (2018) for

future projections under warmer climatic conditions. This implies that changes in the biogeochemical divide could somewhat be similarly impacted from past and future anthropogenic $CO_2$ forcings although the basin specific responses suggest that it may not be a priori simple to predict the pattern or sign of the response of large scale productivity to a given common forcing. Given the economic importance of basin scale productivity and the sensitivity found in past and future similations, reconstructing and understanding the pattern and validating the sign and (model) response of large scale productivity to climate

forcing is therefore critical for assessing not only the sign but also the sensitivity of global productivity to climate change.



The remaining uncertainties include, among others, the use of pre-industrial states for some boundary conditions and the absence of fresh water input, which could modify the spatial response particularly during the early interglacial period which might include the final episodes of continental deglaciation. This is due to the lack of knowledge on such forcing during past climate. Additional model based studies using different Earth system model would be useful to confirm the robustness of our

5  finding and further improve our understanding of the carbon dynamics and the feedback in the ocean under warmer climate. Finally, our model based study suggests that past warm periods experienced considerable carbon cycle and ocean DIC changes, linked to the response of the interior-ocean ventilation and biological productivity to high-latitude warming and interglacial background climate differences. It also suggests that the Atlantic part of the Southern Ocean, which is shown to be a sensitive past climate change, could provide an indicator of future large-scale circulation changes. Close monitoring of the region could

10  be critical in addressing carbon climate feedback in a future warmer climate.

*Data availability.* The full set of model data will be made publicly available through the Norwegian Research Data Archive at https://archive. norstore.no upon acceptance of the paper.

*Competing interests.* The authors declare no competing interests.

*Acknowledgements.* We thank Nadine Goris for her valuable feedback on the first draft of the manuscript. We are grateful to Chuncheng

15  Guo for his technical assistance. This work was supported by the Research Council of Norway funded project THRESHOLDS (254964) and ORGANIC (239965) and Bjerknes Centre for Climate Research project BIGCHANGE. We aknowledge the Norwegian Metacenter for Computational Science and Storage Infrastructure (Notur/Norstore) projects nn2345k, ns2345k, nn1002k, and ns1002k for providing the computing and storing ressources essantial for this study.





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





**Table 1.** Difference in southern sources water ($\triangle_{SSW}$) in the global ocean and for each basin. Row 1 shows the volume ($V_{SSW}$) according to our $PO \geq 0.57$ mol O m$^{-3}$ criteria. Row 2 and 3 summarizes the DIC and water mass age mean value for the two period of study. The changes relative to 115ka are given as a percentage in parenthesis.

|  | Global | Atlantic | Indian | Pacific |
|---|---|---|---|---|
| $\triangle V_{SSW}$ [$10^6 km^3$] | -3.43 (1%) | -18.91 (37%) | +9.76 (8%) | +5.72 (1%) |
| $\triangle DIC_{SSW}$ [$gCm^{-3}$] | -0.34 (1.2%) | -0.92 (3.3%) | -0.61 (2.1%) | -0.36 (1.2%) |
| $\triangle Age_{SSW}$ [years] | -108 (9.3%) | -262 (78.2%) | -152 (29.0%) | -39 (3.0%) |

**Figure 1.** Difference in Sea Surface Temperature ($\triangle SST$) between 125ka and 115ka. Only significant differences (i.e., with absolute value less than the interannual standard deviation over the last 50 years in both 125ka and 115ka) are shown.





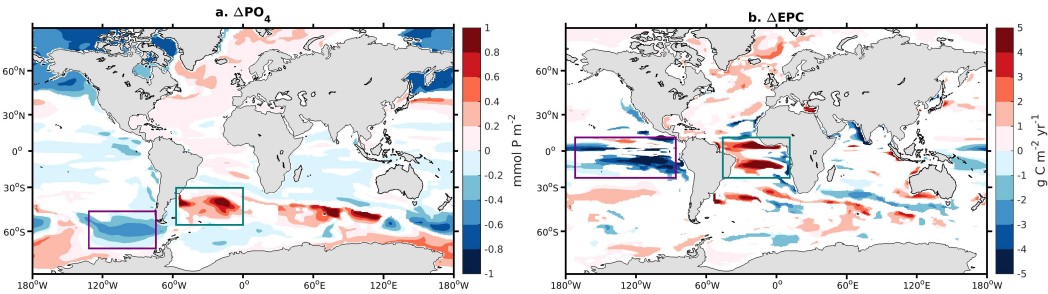

**Figure 2.** Difference in (a) phosphate at the surface ($\triangle PO_4$) and (b) export production of carbon at 100 m depth ($\triangle EPC$) between 125ka and 115ka. When the absolute value of the difference is below the standard deviation over the last 50 years in 125ka and 115ka, the value returns a NaN. Purple and turquoise rectangles highlight the two 'ocean tunnels' linking the sinking/upwelling regions of southern sourced intermediate waters in our model.

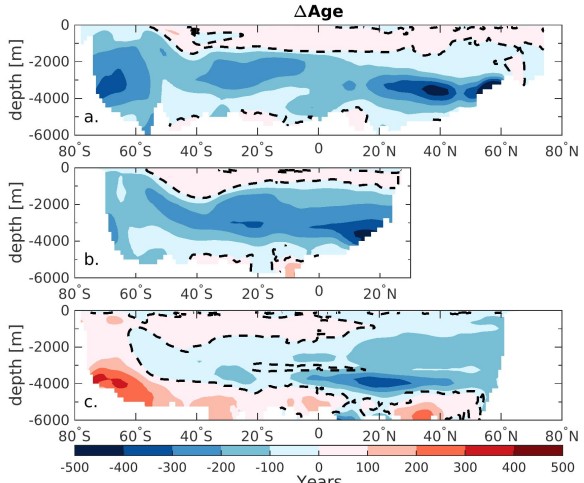

**Figure 3.** Zonally averaged section of the difference in water mass age ($\triangle Age$) between 125ka and 115ka in (a) the Atlantic, (b) Indian and (c) Pacific basins. The dashed-lines display $\triangle Age = 0$





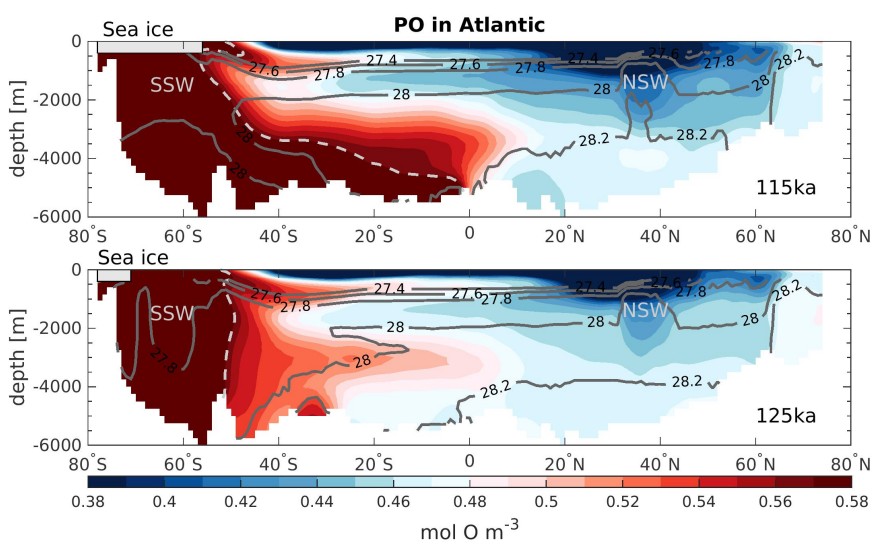

**Figure 4.** Atlantic zonally averaged section of PO as defined by Broecker (1974) in (a) 115ka and (b) 125ka. The light gray dashed-lines delimit the water influenced by the SSW from NSW. The white rectangles represent the sea ice extend during each period and the dark gray solid-lines depict the neutral density.




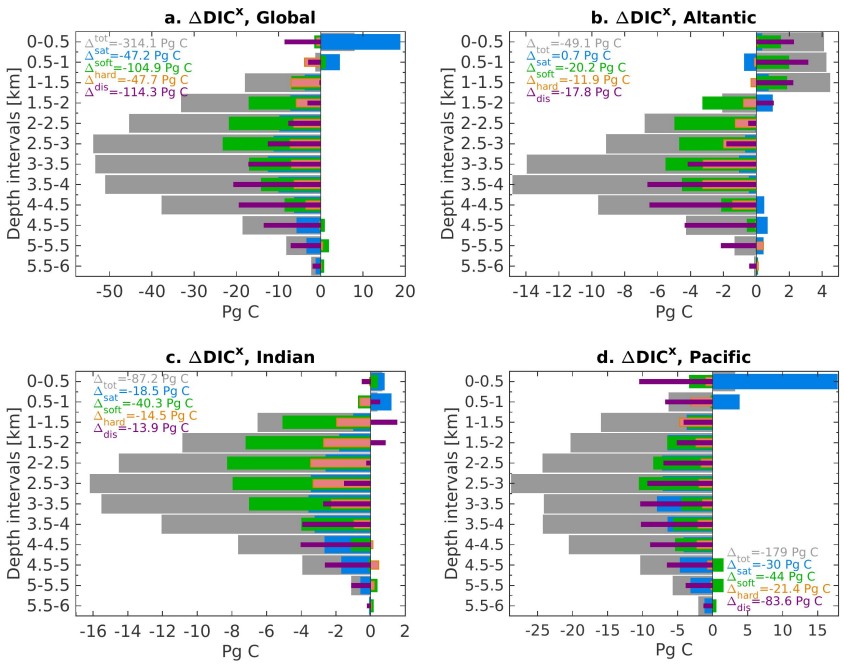

**Figure 5.** DIC differences between 125ka and 115ka ($\triangle DIC^x$) in (a) Global Ocean, (b) Atlantic, (c) Indian and (d) Pacific basins. The $\triangle DIC^x$ is averaged over a 500 m depth interval where 'x' refers to the different components of the DIC. The $DIC^{tot}$ is represented by the gray bars and is decomposed into its 4 components $\triangle DIC^{sat}$ (blue), $\triangle DIC^{soft}$ (green), $\triangle DIC^{carb}$ (orange) and $\triangle DIC^{dis}$ (purple). The sum throughout the water column of each components is given by the legend.





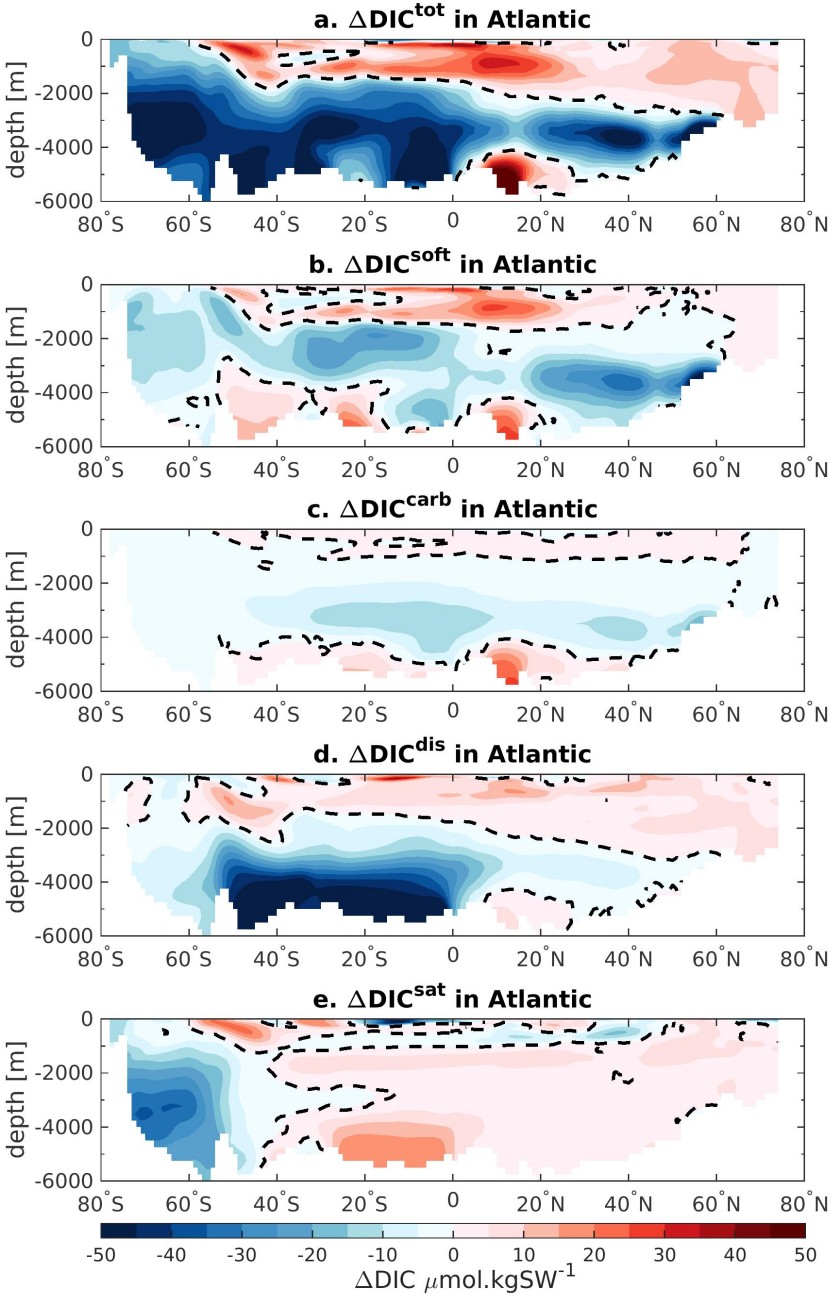

**Figure 6.** Atlantic zonally averaged section of the difference in (a) $\triangle DIC^{tot}$, (b) $\triangle DIC^{soft}$, (c) $\triangle DIC^{carb}$, (d) $\triangle DIC^{dis}$ and (e) $\triangle DIC^{sat}$ between 125ka and 115ka. The black dashed-lines represent the zero values.





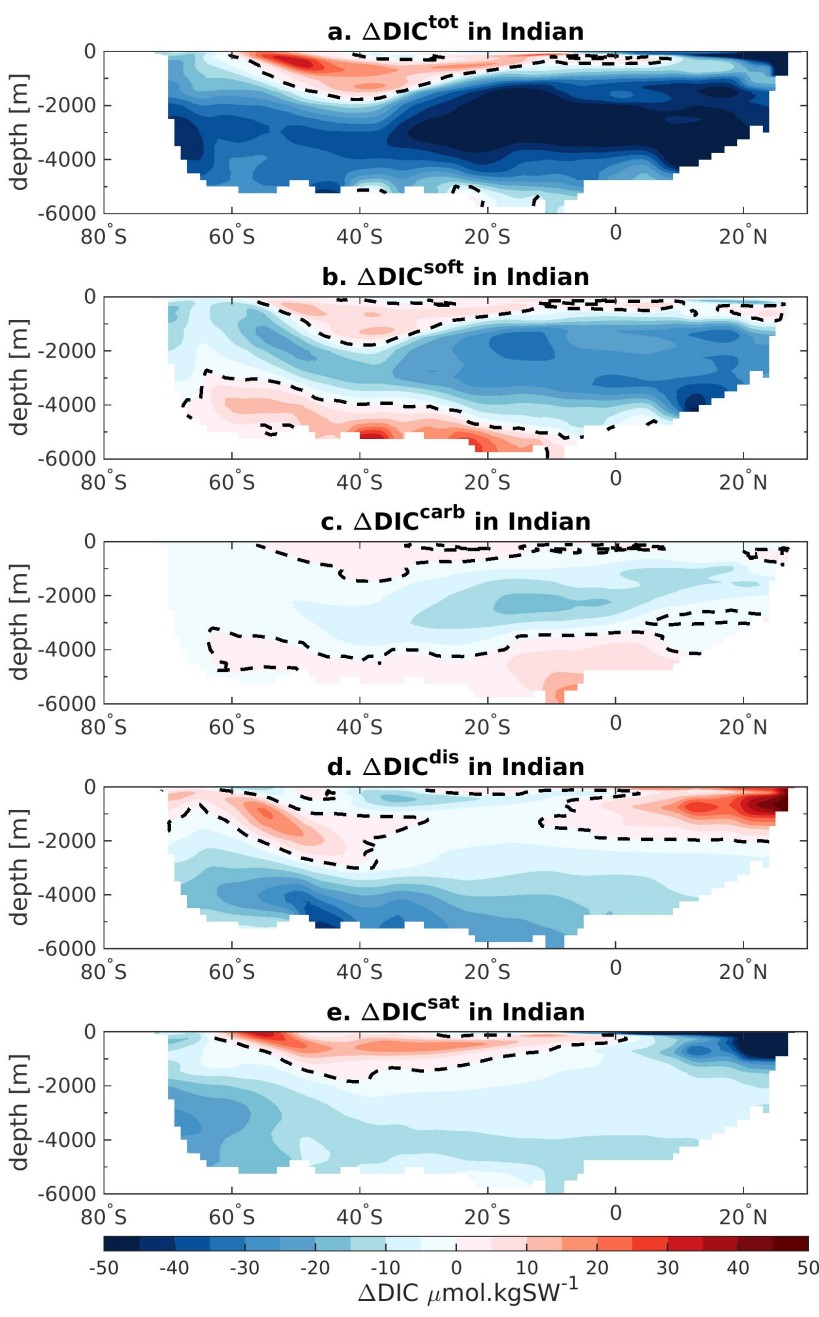

**Figure 7.** Same as Fig. 6 for the Indian Ocean.





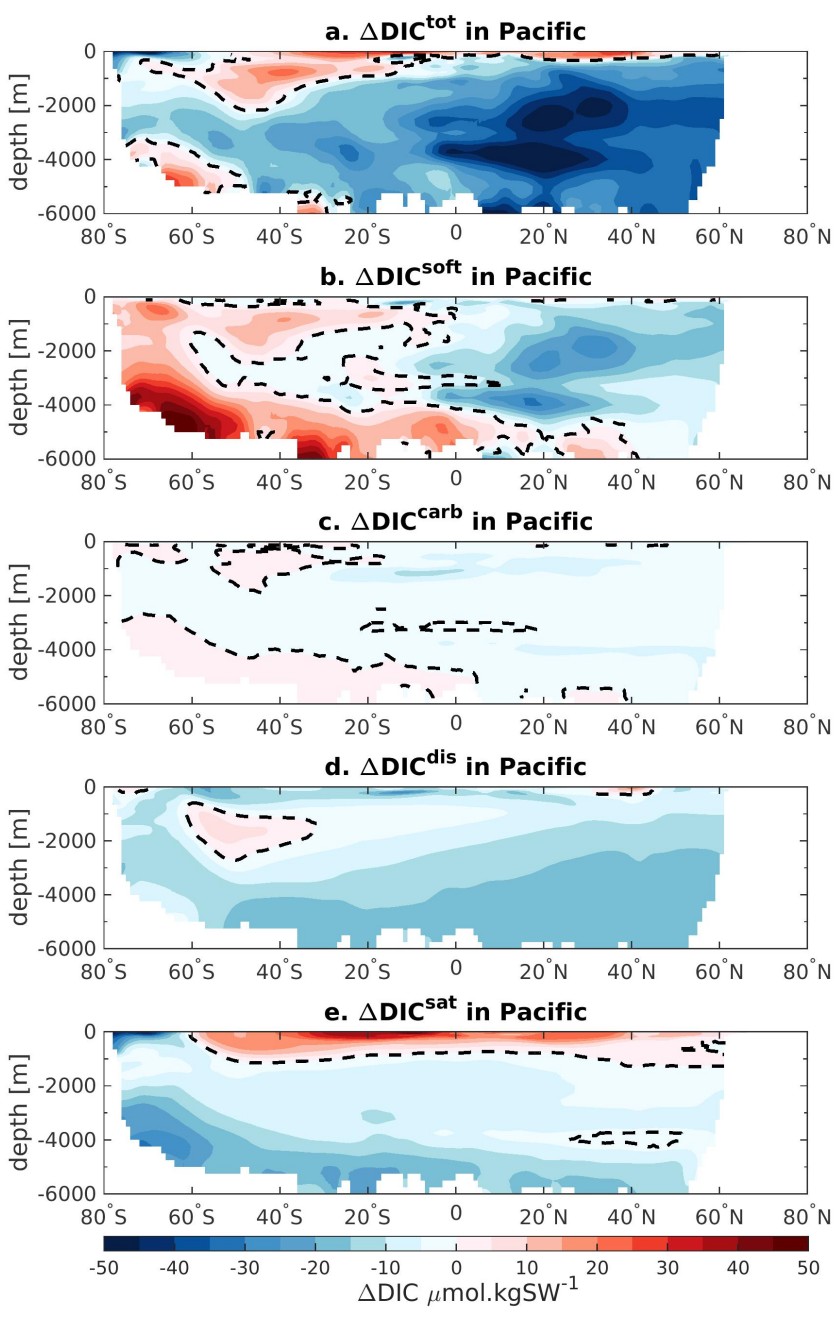

**Figure 8.** Same as Fig. 6 for the Pacific Ocean.





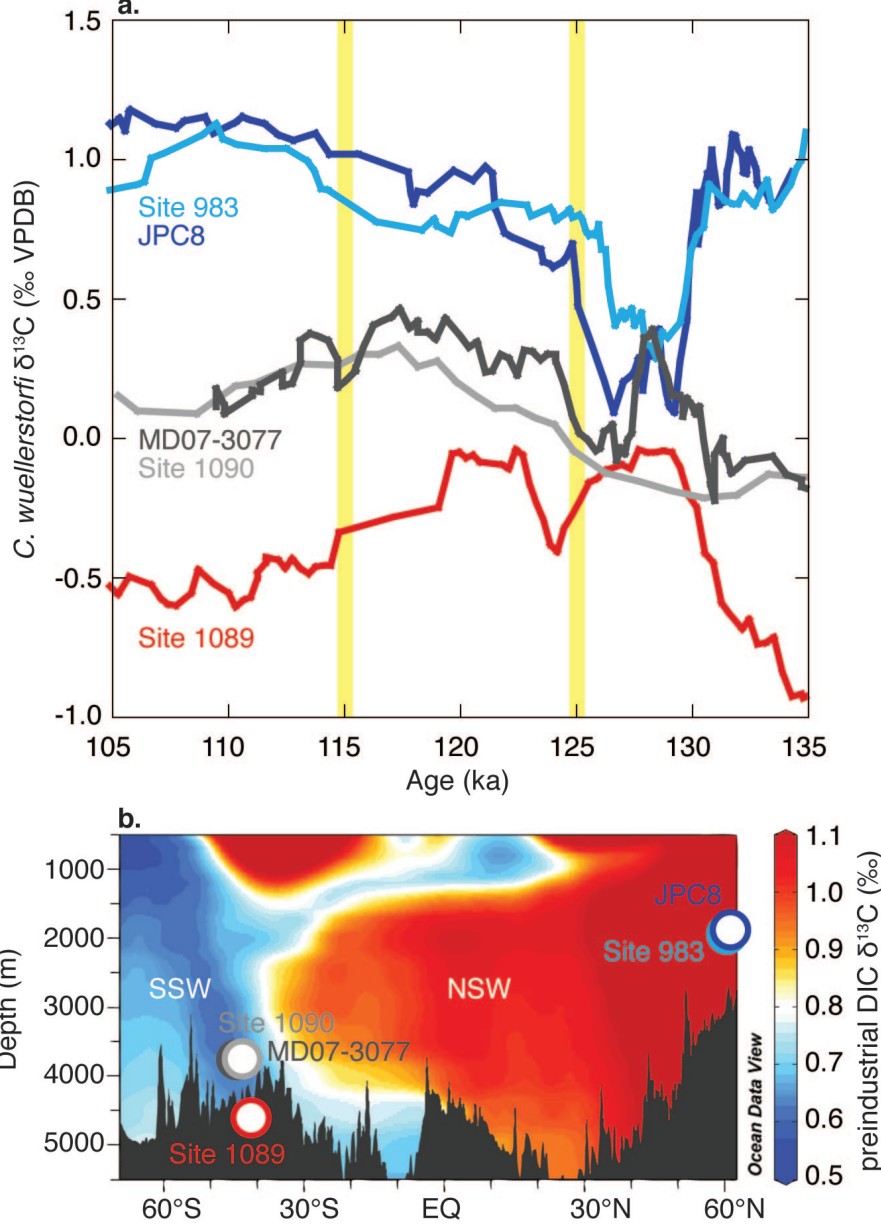

**Figure 9.** (a) LIG reconstructions of bottom water $\delta^{13}C$ based on epibenthic foraminifera Cibicidoides wuellerstorfi (five-point running means) from core sites Ocean Drilling Program (ODP) Site 983 (light blue; Channell et al. (1997)), JPC8 (dark blue; Oppo et al. (1997)), MD07-3077 (dark gray; Gottschalk et al. (2016)), ODP Site 1090 (light gray; Hodell et al. (2003)), and ODP Site 1089 (red; Ninnemann et al. (1999)) plotted versus age (ka) where yellow bands denote the modeled 115ka and 125ka windows. VPDB, Vienna Pee Dee Belemnite. (b) The core locations (Site 983: 60°24′N, 23°38′W, 1984 m depth; JPC8: 61°00′N, 25°00′W, 1917 m depth; MD07-3077: 44°09′S, 14°13′W, 3770 m depth; Site 1090: 42°54′S, 8°54′E; Site 1089: 40°55′S, 9°54′E, 4624 m depth) projected on the preindustrial $\delta^{13}C$ of DIC (Eide et al., 2017) plotted using Ocean Data View (Schlitzer, 2017).   **25**