# Peer review of "Ocean carbon inventory under warmer climate – the case of the Last Interglacial"

_Climate of the Past, 2018_

## Referee Comment (RC1) · Anonymous Referee #1 · 31 Jul 2018

This paper is a modelling analysis of ocean carbon conditions during the Last Inter-glacial, with the aim of serving as an analogous to present-day and future conditions under the effect of human climate change.

The authors do a good job at describing their methods and results, although I believe some extra detail has to be included in some parts of the manuscript, as detailed below. If the authors are able to address these issues then the article could be published by Climate of the Past.

Comments

Page 4, lines 1-8: Very little information is given about how the preformed O2, TALK, and PO4 tracers are calculated by the model. Are they independent from one another?

[Figure]

I think a short (one-two sentences) description should be given.

Page 8, lines 20-25: The experiment on the warmer condition produces lower global DIC. But since atmospheric $CO_2$ is kept constant in each experiment, where is the "missing" carbon of the warmer experiment? Does it get buried by the sediment model? Or does the 125 ka experiment have lower total carbon in all its components? This should be described in the results section.

Page 8, lines starting at 20: I like your detailed analysis of each of the ocean carbon components. I am surprised that you do not link changes in disequilibrium DIC to Southern Ocean sea ice changes. Is this not the case? Please explain briefly in the article.

Page 11, lines starting at 7: The authors interpret their results in terms of d13C in order to compare to sediment core data from the period of study. However, since their model does not directly calculate this tracer, the analysis is subject to misinterpretations. The interpretation of changes in water mass geometry to explain the changes in mid-depth and bottom d13C from sediment cores is plausible. However, is that the only explanation possible? Wouldn't a change in export production affect d13C in a similar way (see Schmittner and Somes, 2016, Paleoceanography)? Also, you talk about the "divergence" between cores JPC8 and Site 983 and Site 1089, it seems that although you are able to explain Site 1089, the increase in d13C in the North Atlantic cores remains unexplained. Do your simulations agree with the two mid-depth North Atlantic cores that show an increasing trend in d13C? If yes, mention in the paper. If not, explain what the missing process could be.

Minor comments:

Title: I think the title should say Last Interglacial, not LIG. I don't think acronyms are good for titles since they leave out people from an outside field to understand the scope of the paper.

Page 7 line 26: There are some typos in the citations.

Page 11 line 11: Replace "an mixture" with "a mixture".

Page 12 line 3: "which corresponds well in absolute magnitude to our findings of -314.1 Pg C". Why is the sign different in Schugers et al. 2006 and your work?

––––––––––––––––––––––––––

---

## Referee Comment (RC2) · Anonymous Referee #2 · 16 Aug 2018

Review of Kessler et al. 2018

General comments

This study aims to investigate an area of research about which we still know relatively little – the behaviour of the ocean carbon cycle during the last interglacial period. The authors do this by comparing two different climate states (125ka and 115ka) in an Earth System Model, contrasting the behaviour of the ocean carbon cycle between these two simulations. This study is a worthy addition to the published literature, but requires some major revisions, particularly in its explanation of the physical mechanisms driving the simulated changes in DIC.

Section 3.2 would benefit from a figure examining changes in Southern Ocean over-

turning strength, looking at both the rate of deep water formation and how the wind-driven circulation changes between the two climate states (strength/ location of the Deacon cell, for example).

I am also surprised that, despite such a large change in SSW/NSW distribution in the Atlantic, the paper does not discuss the Atlantic meridional overturning circulation (AMOC). It is certainly relevant to the conclusions of the study if the two climate states have two different AMOC regimes or if the AMOC is well simulated at all and the impact this has (or doesn't have) on the results.

Specific comments

Please maintain one tense when writing – either past or present, but be consistent.

P2 L19 – The introduction is unusual, in that it spends a lot of time discussing the literature related to terrestrial carbon modelling for the Eemian, rather than focusing on the ocean. I question the relevance of this discussion, particularly as these papers are not discussed further in the context of the results presented within this study.

I would like to see a more robust introduction which discusses not only existing modelling studies of the ocean carbon cycle (eg: Menviel et al. 2012) but also what we might expect to find based on the proxy evidence that is referenced later in the Discussion (consider comparing to Yu et al. 2013). Considering the emphasis of the results on the change in the solubility pump, I would also like to see more written about expected changes in ocean circulation and what previous studies have found (eg. Born et al. 2010; Mokeddem et al. 2014)

Trends in atmospheric CO2 are also discussed, but are not relevant to this study which compares two equilibrated climate states.

P5L24 – This could be a little clearer. Perhaps clarify that DICdis and DICsat equate to preformed carbon, or the solubility pump. This should be noted before these components of the solubility pump are deconstructed and their calculation described.

[Figure]

P6L11 - "We will first describe". Please be more explicit, change to something like "Section 3.1 will describe...". Again, for "The second section", please change to "Section 3.2" and also the sentence beginning with "Finally".

P6L16 - "Simulated sea surface temperature (SST) during the 125ka experiment were warmer globally" with respect to what? Referencing Fig.1 would be appropriate here. I would also recommend a quick comparison to Hoffman et al. 2017 either here or in the discussion comparing the magnitude/ spatial pattern of SST change. This wouldn't require any new figures (particularly as the paper compares to PI SSTs) but simply a qualitative comparison.

P6L16 - "but the changes varied spatially and seasonally, affecting both the ventilation and the nutrient supply at the surface". Please write in the present tense (change varied to vary). I would also like to see contours of $\Delta$max mixed layer depth (or some other metric to diagnose changes in convection/ upwelling) overlaid on Figure 1, or perhaps Figure 2. It would be interesting to see if, with the retreat of Antarctic sea ice in 125ka, the location of deep and intermediate water formation is shifting with respect to 115ka. Based on discussion later in the paper of the Ferrari et al. 2014 sea ice mechanism, it might also be useful to see changes in sea ice extent on these figures and how this is affecting mixed layer/ convective depth and also the location of the formation of these water masses.

P6L18 - "due to a weakening (strengthening) of the mixing process.". Be explicit – do the authors refer to a flattening of the isopycnals resulting in reduced turbulent mixing?

P6L19 - "In the Atlantic, cooler SSTs are simulated during boreal winter and spring ($\Delta$SST $< 0$, Fig. 1a-b), which allow for more upwelled nutrients to the surface (Fig. 2a) via an increasing of the mixed layer depth." - This is not true of the whole basin, be explicit where these changes occur. I would again re-iterate the usefulness of having changes in mixed layer depth/ convective depth contoured on these panels. The grammar of this sentence is also not quite right, please rephrase.

P6L20 - "This higher concentration of nutrients increases the biological production (Fig. 2b) under more favourable warmer blooming season in summer in 125ka ($\Delta$SST > 0, Fig. 1c)." I would note that the authors should refer to 'export production' explicitly, rather than just 'biological production' which may be misunderstood as net primary productivity. Furthermore, the region of the ocean that coincides with increased export production in Fig. 2B is actually cooler in 125ka (Fig. 1c).

P6L23 – I find this sentence to be difficult to understand on first glance. Perhaps rephrase?

P6L26-27 – I would like to see further explanation regarding these 'ocean tunnels', particularly given that our current understanding of ocean dynamics in this region would suggest that intermediate water formed in the South Atlantic sector would take a very indirect route towards the equator, diminishing this signal (flowing north in the Malvinas current, until it is deflected eastward by the Brazil current, before some proportion is entrained into the Benguela Current and reaches the tropics). A reference or two would be sufficient here.

I'm also curious if the authors investigated changes in equatorial/ low latitude winds, as the decrease in export production in the Eastern equatorial Pacific may also be in part due to a weaker Equatorial Undercurrent.

P6L29 – Southern ocean upwelling is largely wind-driven and southern ocean density structure is controlled primarily by salinity rather than temperature. Changes in SSS are mentioned later on P7L30, but it is worth noting here that the change in the Southern Ocean's density structure is not purely due to an increase in SST, which is implied here.

P7L5 - "Despite latitudinally homogeneous forcing..." Not sure what this means? The difference between the two climate states should only be the concentration of greenhouse gases and orbital forcing (which should affect high-latitude insolation). This difference in insolation is zonally homogeneous, but certainly differs with latitude.

P7L32 - 'As a net result, the water mass becomes younger in the SO because of a southward retraction of SSW and southward incursion of more and younger NSW.'. I think this isn't phrased quite correctly. The water mass becomes younger because of a retreat in Antarctic sea ice extent and influx of meltwater, (presumably) an increase in wind-driven upwelling/ mixing and an increase in bouyancy which in turn LEADS to a retraction of SSW.

P8L3 - 'into the interior during the warmest period (Fig 3b).' - please refer to it explicitly as 125ka, or simply add it afterwards (ie: into the interior during the wamer period (125ka – Fig 3b))

P8L3 - 'northward in the Indian, the' - please add Ocean/ Basin.

P8L4 - 'Indian deep water must also decrease (increase)' why is increase included in brackets here? There is no point of comparison.

P8L8 - 'the eastern side waters of the basin are simulated to be older in 125ka by as much as 300 years older.' remove 'older' at the end of the sentence.

P8L8 - 'This older water masses are created in the Pacific SO and are predominantly affected by the strong increase in SST, increasing therefore the stratification' - 'This' should be 'These' and 'increasing therefore the stratification' makes no grammatical sense. Please rephrase.

P8L9 - 'In the northern hemisphere the younger waters are due to to cooler SST (Fig. 1).' I'm sceptical that a slight cooling in the surface sub-tropical and equatorial Pacific would produce a sufficient increase in circulation to decrease the water mass age of the North Pacific basin down past a depth of ∼1000m. I suggest investigating the meridional overturning streamfunction for the Pacific, but more importantly, identifying a mechanism to explain why the deep North Pacific is so much younger. I suspect that, similar to the Southern Ocean, a reorganisation of water masses has simply resulted in this region of the deep ocean to be better ventilated, and has very little to do with

SST in the North Pacific.

P8L16 - 'They are significantly different and basin specific responses of deep water ventilation rates and water mass distribution to uniform changes in forcing.' This sentence is not at all clear (what does 'they' refer to?), please rephrase.

P8L32 - 'The $\Delta$DIC tot of the Indian basin resembles that of the Atlantic at depth, where the soft-tissue and disequilibrium components simulate the strongest decrease (Fig. 5c, green and purple bars). However, the saturation component depicts persistent negative $\Delta$DIC below 1000 m depth, thereby accounting for the second most important component of the decrease throughout the water column.' These two sentences could lead to confusion, as the authors initially state that the Indian Ocean resembles the Atlantic where the soft-tissue pump and disequilibrium components are responsible for the greatest decrease in DIC, however the very next sentence correctly identifies that the saturation component is in fact the second most important component. Please rephrase to enhance clarity.

P9L9 - 'zonally averaged' I'm curious if there is a reason why these plots do not show zonally integrated values of DIC? While I'm sure this would not change the interpretation of the results, it is, in my opinion, the more appropriate depiction of $\Delta$DIC. This is particularly true when each panel should be representing the components of the sum total change in panel a) of each figure.

P9L13 - 'At depth, the changes in SSW and NSW lead to a decrease in younger water masses, hence less remineralized organic matter.' I find this confusing – shouldn't older deep water masses contain more remineralised DIC due to their decreased ventilation rate?

P9L15 - 'Positive change in $\Delta$DIC tot also arises' where?

P9L18 - 'The latter' It is unclear what is being referred to here.

P9L19 - 'The NSW water mass, formed in the North Atlantic, is generally more subject

to biological production during its near surface northward transport before sinking into the interior than the SSW' please provide references to support this claim.

P9L25 - 'due to weaker ventilation induced by stronger SSTs in the Labrador and Nordic sea.' please reference Fig. 1 here. Also, 'stronger' is an inappropriate word here, consider replacing with something like 'warmer'.

P9L28 – Indian *Ocean.

P9L35 - 'Changes in the bottom water DIC soft can be attributed primarily related to the difference in the disequilibrium effect due to stronger carbon export in the Southern Ocean (Fig. 2b) and to a slight decrease in the saturation component. ' Remove 'related'.

P10L10 - 'This lower organic remineralization arises from an increase of the ventilation around 30 âŮę N and a potential increase of the upwelling generating younger water masses (Fig. 3c)' This sentence is misleading – there isn't increased ventilation at 30N, rather, water masses at 30N are better ventilated (likely via some pathway that results in upwelling in the Southern Ocean). Please rephrase to avoid confusion. Also, if I understand this correctly, ' This lower organic remineralization' refers to the decrease in DICsoft rather than an actual decrease in the remineralization rate in the ocean. If this is the case, consider rephrasing this to something along the lines of 'this decrease in biogenic carbon ...'.

P10L11 - 'This is in good agreement with the increased carbon export production (Fig. 2b)' But export production is largely unchanged or negative in the Pacific Ocean according to Fig. 2b.

', inducing positive $\Delta DIC_{tot}$ near the surface, and with the cooler SST depicted in Fig. 1' Negative SSTs are only evident in Boreal winter and spring – does this outweigh the effect of considerably warmer SSTs in summer and autumn?

P10L21 - 'This SSW mainly composes the Pacific interior ocean.' please rephrase.

P10L30 - 'Significant decreases of the ocean carbon' replace 'of the' with 'in'.

P10L31 – This is a good summary, but I would suggest that instead of saying 'Most of this decrease', give the exact %.

I would also like to see an extra sentence at the end of this summary paragraph highlighting the mechanisms responsible for the change in ocean ventilation. This could include AMOC strength, changes in SSW/NSW, changes in AABW/IW formation, sea ice changes etc.

Comparison to proxies regarding water masses is good, though I would suggest one or two extra studies that could be referenced during the discussion (please see list at the end of this document).

P11L24 - 'A sense of the scale of the changes model simulations can be gained through comparison to previous modelling efforts where atmospheric CO 2 was not fixed.' this sentence does not make sense, please rephrase.

I'd also like to see comparisons to Menviel et al. 2012 here.

P12L1 - 'obtained a difference in atmospheric CO 2 concentration and terrestrial carbon storage of about 40 PgC and 350 PgC, respectively' If you refer to an atmospheric concentration, then please change the 40 PgC to ppm equivalent, or alternatively, rephrase to refer to "atmospheric and terrestrial carbon storage" instead. The authors should also be clearer here: is this +350Pg and -40Pg? Similarly, the following line claims that a change in ocean carbon of 310 Pg corresponds well to your findings of -314.1 PgC. These values are opposite, is the former supposed to be -310?

P12L4 - 'However, the changes in CO 2 concentration in the atmosphere that they simulated steadily increases' This is poorly worded and gives impression that the rate of change of atmospheric CO2 is varying somehow. Please rephrase. Something along the lines of "However, their simulated atmospheric CO2 steadily increases over this period".

P12L31 - ' The ongoing global warming raises questions about the oceanic carbon sink and its efficiency under a warmer climate condition.' Be specific. 'Ongoing anthropogenic warming...' and remove 'The' from the start of the sentence.

P12L32 - 'In this study, we use a fully-coupled NorESM' please change 'a' to 'the' (unless of course there is more than one version of Nor-ESM, which should then be stated).

P13L1 - 'We focus on the differences that occurred in 125ka in comparison to 115ka, specifically the differences at global and basin scales.' This is vague – be explicit that you're looking at changes in the ocean carbon cycle.

P13L2 - 'Thereby, to our knowledge, it is the first attempt in elucidating the biogeochemical and physical processes that are responsible for the ocean carbon inventory changes under warmer climate conditions during the LIG.' Please see Menviel et al. 2012.

P13L24 - 'The western Pacific is influenced by water masses coming from the Pacific Southern Ocean, with a warmer SST that hinders the ventilation and increases the residence time of the interior water masses on the eastern side of the basin.' On P8L6, the authors state that the Western Pacific is ventilated by water originating from the Atlantic. Either this section or that in section 3.2 needs to be clarified/ corrected.

Minor typographical errors/ suggestions

Please capitalise names of basins/ hemispheres throughout the manuscript.

P1L7 - "We find a considerable weaker ocean dissolved..." this sentence does not make grammatical sense. I would suggest changing it to "We find considerably reduced ocean dissolved..."

L14 - "...restricting the extend of DIC rich southern sourced water reducing the storage of biological remineralized carbon at depth." Typo. 'extend' should be 'extent'.

[Figure]

Furthermore, consider rephrasing as follows "...restricting the extent of DIC rich southern sourced water, thereby reducing the storage of biological remineralized carbon at depth.

P1L20 - "If the anthropogenic greenhouse..." remove 'the'.

P1L23 – consider rephrasing "For this, the changes in the the warm Eemian period may be considered as an analog for a future warmer climate." to "The changes in the the warm Eemian period may therefore be considered an analog for a future warmer climate.

P3L26 – prescribed should be prescribes.

P3L27 - "(>100m depth)" should this not be <100m depth? Or are you referring to the sinking rate of the particles that have left the euphotic zone?

P3L31 - 'vertical advection': convection?

P4L24 - 'the dissolved inorganic carbon'. 'the' is not necessary.

P5L25 – Remove 'mostly' - vague language.

P6L13 - 'Each analysis have' - please rephrase.

P7L2 - 'southern hemisphere' should be Southern Hemisphere.

P7L17 - 'translates to a stronger ventilation rate age and' remove 'age'.

P7L22 - 'since the SST in the SO is rather warmer in 125ka' remove 'rather'.

P7L26 - 'byFerrari et al. (2014)' space missing

P7L26 - 'study,Luo et al...' space missing.

P8L27 - 'main contributors for the weaker carbon inventory' replace weaker with reduced.

P9L3 - 'simulating a strong positive difference of about +18 PgC' strong with respect to

what? I would suggest removing such descriptive words.

P9L6 - 'However, the saturation component has also a considerable influence' 'has also' should be 'also has'.

P9L29 - 'Only in the region that may correspond to the AAIW the simulated 4DIC tot are positive (Fig. 7a, red shade).' Poor grammar, please rephrase.

P11L6 - 'has been previously been inferred' remove second 'been'.

P11L27 - 'but at slightly weaker decrease than in our study.' grammatically incorrect. Please rephrase.

P12L8 - 'our model study shows an heterogeneous response' 'a' not 'an'.

P12L10 - 'Such heterogeneous response of the biogeochemical divide have also been highlighted by Moore et al. (2018) for future projections under warmer climatic conditions.' Consider rephrasing to 'Such a response...' and '...has also been highlighted...'

P12L11 - 'This implies that changes in the biogeochemical divide could somewhat be similarly impacted from past and future anthropogenic CO 2 forcings' Poorly phrased, please correct.

P12L12 - "Reconstruct and understand the pattern and signs of past responses of large scale productivity to climate forcing are therefore critical for assessing not only the sign but also the sensitivity of different regions to climate change." Poorly phrased, please re-write.

P12L15 - 'Factors that could influence ocean carbon storage including sea level, riverine input of nutrients, and atmospheric dust loading, which are all set to preindustrial levels in our simulations, but may have been different in the LIG.' remove 'which', otherwise sentence does not make grammatical sense.

P12L23 - 'Our quasi-equilibrated model simulations for 115ka and 125ka, also lack ice sheet and the corresponding freshwater input variability, do not address such shorterterm changes that could affect the ocean carbon inventory (Stocker and Schmittner, 1997).' *and do not address.

P13L32 - 'a priori' typically written in italics, though this is only a suggestion.

References

Few references include a doi. Please add them wherever available.

Table 1: 'southern sources water' correct to 'sourced' I'm also curious as to why values for NSW were not also included in this table and discussed in the manuscript.

Figure 2. The authors refer to this box as a 'turquoise rectangle' here but as a green rectangle elsewhere in the manuscript. Please be consistent and change one or the other.

Figure 4. Please include units for the neutral density contours in the caption.

Figure 5. Again, I question the utility of averaging over 500m intervals rather than summing. Particularly when dealing with bulk differences. Is there a reason for this?

Recommended references

Menviel et al. 2012 - Simulating atmospheric CO2, 13C and the marine carbon cycle during the Last Glacial–Interglacial cycle: possible role for a deepening of the mean remineralization depth and an increase in the oceanic nutrient inventory

Yu et al. 2013 - Responses of the deep ocean carbonate system to carbon reorganization during the Last Glacial–interglacial cycle

Born et al. 2010 Sea ice induced changes in ocean circulation during the Eemian

Born et al. 2011 Late Eemian warming in the Nordic Seas as seen in proxy data and climate models

Hoffman et al. 2017 Regional and global sea-surface temperatures during the last interglaciation

Mokeddem et al. 2014 Oceanographic dynamics and the end of the last interglacial in the subpolar North Atlantic

---

## Referee Comment (RC3) · V. Brovkin (Referee) · 24 Aug 2018

Kessler et al. presented analysis of two time-slice experiments for the Marine Isotope Stage 5, comparing warm periods of 125 ka and onset of glaciation at 115 ka using Nor-ESM. The paper is focused on changes in DIC components in the ocean, and provide a novel way to separate DIC into components associated with different physical and biological mechanisms. I found it quite useful for understanding of biogeochemistry changes in paleo simulations. An obvious caveat is that the simulations are not transient, but this is a usual limitation of state-of-the art ESMs. The paper is well written, and main findings are well described in the manuscript. I have few comments/suggestions aimed at more detailed understanding of the method and model results.

Method:

- Are DIC changes due to changes in SSTs (solubility) included into DeltaDIC^sat, DeltaDIC^dis, or both? Please discuss this, as SST changes are essential for differences between cold and warm states.

- My understanding is that the biological pump changes are caused by changes in the physical circulation. If it is so, could it be briefly mentioned?

- Differences in atmospheric CO2 and total alkalinity between two simulations are minor (p. 4), but ocean DIC changes are minor too (< 1% of ocean DIC content). DIC effect of changes in these boundary conditions (pCO2, TALK) needs to be discussed.

- Please provide differences in weathering/sedimentation between two simulations, as they are essential for carbonate chemistry.

Results:

- One of the interesting findings is that the ocean circulation, in particular formation of deep waters in the Southern ocean, is very sensitive to the orbital forcing changes via SH sea ice changes. Please provide 2-D plots for extent of summer/winter sea ice in the Southern Ocean. Plots of meridional overturning in Atlantic might be insightful too.

- Figure 4. What is the threshold in PO separating the SSW from NSW (dashed lines)? Is it the same value for both plots? If total phosphate inventory is different in two simulations, should PO be corrected for it?

- Figure 9. A relevance of this figure to the study is unclear.

---

## Author Comment (AC1) · 16 Oct 2018

We thank the anonymous referee#1 for his/her constructive comments which help to improve our paper. Below, we reply to the comments made by the referee#1 point by point. The responses are constructed as follows (1) original comments from the referee in **bold**, (2) our response in *italics*, and (3) description of changes applied in the revised manuscript in blue.

**Page 4, lines 1-8: Very little information is given about how the preformed O2, TALK, and PO4 tracers are calculated by the model. Are they independent from one another? I think a short (one-two sentences) description should be given.**

*Response: We have added the following statement.*

Revision in Sect. 2.1: "At surface, these preformed tracers are set to their non-preformed value and are advected passively by the ocean circulation in the interior without any other sources and sinks."

**Page 8, lines 20-25: The experiment on the warmer condition produces lower global DIC. But since atmospheric CO2 is kept constant in each experiment, where is the "missing" carbon of the warmer experiment? Does it get buried by the sediment model? Or does the 125 ka experiment have lower total carbon in all its components? This should be described in the results section.**

*Response: The difference in the sediment carbon is about $6.10^{-4}$ PgC between 125ka and 115ka, which is negligible compared to the difference in $DIC^{tot}$ (314 Pg C). The equilibrium simulations with fixed GHG and orbital forcings performed in our study are based on standard protocols (e.g. PMIP-paleoclimate Model Intercomparison Project) to study the climate or ocean circulation responses at certain periods in the past (Weber et al., 2007; Van Meerbeeck et al., 2009). When the ocean carbon cycle is included in such simulations, as in our study, it is implicitly assumed that the residual of atmosphere and ocean carbon budgets is offset by changes in the terrestrial reservoir.*

Revision in Sect. 3.4: "Since the atmospheric CO2 is kept constant in each experiment, this carbon loss in 125ka compared to 115ka is implicitly assumed to be balanced out by the changes in the land carbon reservoir. "

**Page 8, lines starting at 20: I like your detailed analysis of each of the ocean carbon components. I am surprised that you do not link changes in disequilibrium DIC to Southern Ocean sea ice changes. Is this not the case? Please explain briefly in the article.**

Response: Changes in DICdis is complex as it can be affected by many different factors such as changes in the physical pump, overturning circulation or the biological pump. No general process could be attributed to its variation which seems to be regionally affected. In the Atlantic bottom water it is mainly driven by the sea-ice-induced NSW/SSW water masses re-organization. However, near surface changes remain relatively unexplained.

Revision in Sect. 5: However, the processes affecting the disequilibrium component can arise from

different factors such as changes in the physical pump, overturning circulation or biological pump. No general process could be attributed to its variation which seems to be regionally affected. While the SSW seem to become more undersaturated in 125ka, the NSW seem to be more saturated. Further experiments with for instance fixed biological productivity or overturning circulation could help to identify the sensitivity of this component to such factors, but remain too expensive to perform with our model.

**Page 11, lines starting at 7: The authors interpret their results in terms of d13C in order to compare to sediment core data from the period of study. However, since their model does not directly calculate this tracer, the analysis is subject to misinterpretations. The interpretation of changes in water mass geometry to explain the changes in mid-depth and bottom d13C from sediment cores is plausible. However, is that the only explanation possible? Wouldn't a change in export production affect d13C in a similar way (see Schmittner and Somes, 2016, Paleoceanography)?**

*Response: Indeed, bottom water $\delta^{13}C$ can be influenced by several factors including changes to export production and local nutrient cycling. However, it is unlikely that export productivity changes alone caused the mid-depth and bottom $\delta^{13}C$ changes. Export productivity reconstructions from the Site 1089 region show no discernible change over the time interval when Site 1089 bottom water $\delta^{13}C$ decreased (e.g., Martínez-Garcia et al., 2014, Science), while the magnitude of the Site 1089 bottom water $\delta^{13}C$ decrease (~0.4‰) indicates a considerable nutrient concentration increase (following scaling in e.g. Eide et al., 2017, Global Biogeochemical Cycles). The character and spatial pattern of bottom water $\delta^{13}C$ changes in the wider South Atlantic region also indicate that, on millennial timescales, bottom water $\delta^{13}C$ variability in this region is most easily explained by water mass geometry changes (Oppo and Fairbanks, 1987, Earth and Planetary Science Letters; Charles and Fairbanks, 1992, Nature; Ninnemann et al., 1999, Global and Planetary Change), including for the LIG time interval we discuss in the manuscript (Govin et al., 2009, Paleoceanography). For example, they occur independently of the (core-specific) surface ocean nutrient regime (Ninnemann & Charles, 2002, Earth and Planetary Science Letters).*

*After careful consideration of all referees suggestions, we shortened the section describing precisely the changes in $\delta^{13}C$ and refer exclusively to the literature instead. Therefore, we also removed the previous Fig. 9 in order to keep our study clear and avoid confusion.*

Revision in Sect. 4: "Similar to our results, expanded SSW in the late compared to early LIG is inferred from such reconstructions to explain bottom water $\delta^{13}C$ decreases in different regions proximal to the Southern Ocean (Govin et al., 2009; Ninnemann et al., 2002). Bottom water $\delta^{13}C$ reconstructions indicate less influence of NSW in the deep South Atlantic at 115ka than at 125ka (Ninnemann et al., 1999; Govin et al., 2009)."

**Also, you talk about the "divergence" between cores JPC8 and Site 983 and Site 1089, it seems**

**that although you are able to explain Site 1089, the increase in d13C in the North Atlantic cores remains unexplained. Do your simulations agree with the two mid-depth North Atlantic cores that show an increasing trend in d13C? If yes, mention in the paper. If not, explain what the missing process could be.**

*Response: The increasing $\delta^{13}C$ in the mid-depth North Atlantic most likely reflects an increase in the preformed $\delta^{13}C$ (Fronval et al., 1998). This increase cannot be clearly assessed with our model as it does not include $\delta^{13}C$, for example. However, persistent vigorous AMOC ventilating the mid-depth Atlantic Ocean is simulated in both experiments, consistent with proxy reconstructions. Following suggestions by referees #1 and #3, we have added more references to both proxy and model studies showing this.*

Revision in Sect. 4: "In addition, persistent (millennial-scale) mid-depth NSW ventilation extending from the LIG and into the subsequent glacial inception is also suggested based on proxy reconstructions (McManus et al., 2002; Mokeddem et al., 2014) and is consistent with model simulations (Born et al., 2011; Wang et al., 2002). In our study, even though the AMOC is simulated slightly stronger and deeper in 125ka, vigorous AMOC persists during 115ka ventilating the North Atlantic mid-depth. We also note that our model may not properly represent the North Atlantic overflows due to its sparse resolution. This can further add uncertainties to North Atlantic water ventilation. "

References:

Born A., Nisancioglu K. H., Risebrobakken B.: Late Eemian warming in the Nordic Seas as seen in proxy data and climate models, Paleoceanography, 26(2), PA2207, 2011. https://doi.org/10.1073/pnas.1322103111

Charles, C.D. And Fairbanks, R.G.: Evidence from Southern Ocean sediments for the effect of the North Atlantic deep water flux on climate, Nature, 355, 416-419, 1992.

Eide M., Olsen A., Ninnemann U.S. and Johannessen T.: A global ocean climatology of preindustrial and modern ocean 13C, Global Biogeochem. Cycles, 31, 2017. https://doi.org/10.1002/2016GB005473

Fronval, T., Jansen E., Haflidason H., Sejrup H. P.: Variability in surface and deep water conditions in the Nordic Seas during the last interglacial period, 17(9-10), 963-985, 1998. https://doi.org/10.1016/S0277-3791(98)00038-9

Govin, A., E. Michel, L. Labeyrie, C.Waelbroeck, F. Dewilde, and E. Jansen (2009), Evidence for northward expansion of Antarctic Bottom Water mass in the Southern Ocean during the last glacial inception, Paleoceanography, 24, PA1202, https://doi.org/10.1029/2008PA001603

McManus J. F., Oppo D. W., Keigwin L. D., Cullen J. L., Bond G. C.: Thermohaline circulation and

prolonged interglacial warmth in the North Atlantic, Quat. Res., 58(1), 17–21, 2002. https://doi.org/10.1006/qres.2002.2367

Martinez-Garcia, A., Sigman, D.M., Ren, H., Anderson, R.F., Straub, M., Hodell, D.A., Jaccard, S.L., Eglinton, T.I. And Haug, G.H: Iron fertilization of the Subantarctic Ocean during the Last Ice Age, Science, 343, 1347-1350, 2014. https://doi.org/10.1126/science.1246848

Mokeddem Z., McManus J. F., Oppo D. W.: Oceanographic dynamics and the end of the last interglacial in the subpolar North Atlantic, Proceedings of the National Academy of Sciences of the United States of America, 111(31), 11253-11268, 2014. https://doi.org/10.1073/pnas.1322103111

Ninnemann U. S., Charles C. D., and Hodell D. A.: Origin of global millennial scale climate events: constraints from the Southern Ocean deep sea sedimentary record, Geophysical Monograph-American Geophysical Union, 112, 99-112, 1999.

Ninnemann U. S., Charles C. D.: Changes in the mode of Southern Ocean circulation over the last glacial cycle revealed by foraminiferal stable isotopic variability, Earth and Planetary Science Letters, 201(2), 383-396, 2002. https://doi.org/10.1016/S0012-821X(02)00708-2

Oppo D. W., Fairbanks R. G.: Variability in the deep and intermediate water circulation of the Atlantic Ocean during the past 25,000 years: Northern Hemisphere modulation of the Southern Ocean, Earth and Planetary Science Letters, 86(1), 1-15, 1987. https://doi.org/10.1016/0012-821X(87)90183-X

Wang Z., Mysak L. A.: Simulation of the last glacial inception and rapid ice sheet growth in the McGill Paleoclimate Model, Geophys. Res. Lett., 29(23), 2102, 2002. https://doi.org/10.1029/2002GL015120

Weber, S. L., Drijfhout, S. S., Abe-Ouchi, A., Crucifix, M., Eby, M., Ganopolski, A., Murakami, S., Otto-Bliesner, B., and Peltier, W. R.: The modern and glacial overturning circulation in the Atlantic ocean in PMIP coupled model simulations, Clim. Past, 3, 51-64, 2007. https://doi.org/10.5194/cp-3-51-2007

---

## Author Comment (AC2) · 16 Oct 2018

We thank V. Brovkin for his constructive comments which further improve our paper. Below, we reply to the comments made by V. Brovkin point by point. The responses are constructed as follows (1) original comments from the referee in **bold**, (2) our response in *italics*, and (3) description of changes applied in the revised manuscript in blue.

**- Are DIC changes due to changes in SSTs (solubility) included into DeltaDIC^sat, DeltaDIC^dis, or both? Please discuss this, as SST changes are essential for differences between cold and warm states.**

*Reponse: DICsat is computed with CO2SYS using the boundary conditions from each experiment (Sect. 2.3). Thereby, changes in DICsat include all the parameters listed in Sect. 2.3 and among them preformed ALK, SSTs and atmospheric CO2 concentration. However, in our model the changes in preformed alkalinity seems to mainly control the changes in DICsat. DICdis is the residual member of the equation and is obtained by subtracting all DIC components to the total DIC (DICtot). Thus, changes in DICdis also include changes in SSTs.*

Revision in Sect. 2.3 after Eq. 2: "In our model it seems to mainly refer to changes in preformed ALK. However, we compute this variable offline with the inorganic carbon chemistry program CO2SYS developed in Matlab (van Heuven et al., 2011) computation including other parameters such as the model output of preformed alkalinity ($TALK^{pre}$), preformed phosphate ($PO4^{pre}$), surface silicate, salinity and temperature. In addition, the atmospheric $CO_2$ concentration from each experiment is used."

**- My understanding is that the biological pump changes are caused by changes in the physical circulation. If it is so, could it be briefly mentioned?**

*Response: Yes, V. Brovkin is right, the biological pump changes are mainly driven by changes in interior ocean ventilation timescale.*

Revision in the abstract: "The biological pump is mainly driven by changes in interior ocean ventilation timescales, but the processes controlling the changes in ocean DIC disequilibrium remain difficult to assess and seem more regionally affected. While Atlantic bottom water disequilibrium is affected by the sea-ice induced SSW/NSW organization, the upper layer changes remain unexplained."

**- Differences in atmospheric CO2 and total alkalinity between two simulations are minor (p. 4), but ocean DIC changes are minor too (< 1% of ocean DIC content). DIC effect of changes in these boundary conditions (pCO2, TALK) needs to be discussed.**

*Response: The boundary conditions on pCO2 and TALK only cause a change in DIC of about 32 PgC which remains small compared to the 314 PgC of changes between the two periods by taking all components of the carbon cycle into account. This infers that other processes are controlling the DIC inventory in the Ocean.*

Revision in Sect. 2.2: "The differences in pCO$_2$ and TALK budget are small between the two experiment. Such changes would affect the DIC budget of about 32 PgC."

**- Please provide differences in weathering/sedimentation between two simulations, as they are essential for carbonate chemistry.**

*Reponse: V. Brovkin is correct in that weathering fluxes and sedimentation rates play an important role into the carbonate chemistry for long time scale simulations. In our simulations the difference in sedimentation rate between 125ka and 115ka is about 6.10$^{-4}$ PgC, which is negligible compared to the change in DIC in the water column and therefore has only a minor effect on the carbon chemistry. The model we used for this study does not include weathering fluxes in its module.*

Revision in Sect. 2.2: "In addition, the difference in sedimentation rates between the two experiments (≈6.10$^{-4}$ PgC) appears to be negligible compared to the difference in DIC budget."

Revision in Sect. 4: "In addition, our model does not include weathering fluxes, which might influence the carbon budget on such longtime scale."

**- One of the interesting findings is that the ocean circulation, in particular formation of deep waters in the Southern ocean, is very sensitive to the orbital forcing changes via SH sea ice changes. Please provide 2-D plots for extent of summer/winter sea ice in the Southern Ocean. Plots of meridional overturning in Atlantic might be insightful too.**

*Response: We concur. We have added two figures (Fig. 1 and Fig. 2). Figure 1 depicts in shade the differences in SSTs (125ka – 115ka), the sea ice extent as contour lines (115ka in green and 125ka in purple) and the changes in the mixed layer depth as black (+100m depth) and blue (-100m depth) lines. We have added a completely new subsection (Sect. 3.1) that describes specifically the changes depicted in Fig. 1. Figure 2 shows the changes in the overturning circulation in the Southern Ocean during the two experiments and is explained in Sect. 3.2.*

**- Figure 4. What is the threshold in PO separating the SSW from NSW (dashed lines)? Is it the same value for both plots? If total phosphate inventory is different in two simulations, should PO be corrected for it?**

*Response: Yes, we used the same value in both experiment to separate SSW by NSW (PO = 0.57 mol O m$^{-3}$). V. Brovkin is correct in that a change in phosphate inventory would affect this threshold. However, the phosphate inventory between 115ka and 125ka experiments differs from 4.6%, which remains relatively small.*

*Figure R1 shows the distribution of PO at the surface of the ocean. In both experiments, SSW and NSW can be clearly identified in the Southern Ocean. Assuming threshold of about 0.57 mol O m-3 corresponds relatively well to the limit depicted by the surface PO for both experiments.*

[Figure]

*Fig R1: Surface PO in 115ka (left) and 125ka (right) experiments.*

**- Figure 9. A relevance of this figure to the study is unclear.**

*Response: We agree. After careful consideration of all referee comments, we removed the previous Fig. 9 to keep this study clear and focus the reader on the model results. We still include a short comparison to proxy reconstruction, but only refer to the literature.*

Reference:

*van Heuven, S., Pierrot, D., Rae, J. W. B., Lewis, E., and Wallace, D. W. R.: Matlab program developed for CO2 system calculations.* ornl/cdiac-105b., Carbon Dioxide Information Analysis Center, Oak Ridge National Laboratory, US Department of Energy, Oak Ridge, Tennessee, http://cdiac.ornl.gov/ftp/co2sys/CO2SYS_calc_MATLAB_v1.1/, 2011.

---

## Author Comment (AC3) · 16 Oct 2018

We thank Referee#2 for his/her positive and constructive feedback on our study and for the new references he/she provided. In the revised manuscript, we have addressed most or all of the comments raised and we think that the manuscript has been significantly improved since. Below, please find point by point response to each of Referee#2 comment. They are constructed as follows (1) original comments from the referee in **bold**, (2) our response in *italics*, and (3) description of changes applied in the revised manuscript in blue.

**Section 3.2 would benefit from a figure examining changes in Southern Ocean overturning strength, looking at both the rate of deep water formation and how the wind driven circulation changes between the two climate states (strength/ location of the Deacon cell, for example). I am also surprised that, despite such a large change in SSW/NSW distribution in the Atlantic, the paper does not discuss the Atlantic meridional overturning circulation (AMOC). It is certainly relevant to the conclusions of the study if the two climate states have two different AMOC regimes or if the AMOC is well simulated at all and the impact this has (or doesn't have) on the results.**

*Response: We follow the referee#2 suggestion and have added a new figure of Southern Ocean overturning strength during both experiment so as a short analysis. The northern structure of the AMOC and the analysis of the wind during these two period have been documented in Luo et al. (2018) and are not repeated in this study. However, few statements have been added in the subsection 3.2.*

Revision in Sect. 3.2:

"In order to analyze the water mass properties, it is useful to examine the changes in the overturning circulation. Figure 2 shows the global overturning stream function in the Southern Ocean for both experiments. The Antarctic circumpolar current is simulated stronger and deeper in 125ka compared to 115ka (Fig. 2). This strengthening is mainly occurring in the Pacific section of the Southern Ocean (near 50°S), suggesting an increase of the ventilation rate of the intermediate waters formed in this region. The Atlantic section of the Southern Ocean remains weakly modified. Indeed, using the same model simulations as the present study, Luo et al. (2018, supplementary information Fig. S8) show that the surface wind speed in the east and west southern Atlantic are relatively similar in 125ka and 115ka. In addition, they also show that the simulated AMOC in 125ka is as vigorous as in 115ka but deepen by about 300m depth. This suggests that the mid-depth and bottom water in the North Atlantic Ocean should be better ventilated in 125ka than in 115ka."

[Figure]

**Figure 2:** *Section of the Souther Ocean Overturning stream function in 115ka (top panel) and 125ka (bottom panel).*

**Please maintain one tense when writing – either past or present, but be consistent.**

*Response: We concur and corrections have been made throughout the text.*

**P2 L19 – The introduction is unusual, in that it spends a lot of time discussing the literature related to terrestrial carbon modelling for the Eemian, rather than focusing on the ocean. I question the relevance of this discussion, particularly as these papers are not discussed further in the context of the results presented within this study. I would like to see a more robust introduction which discusses not only existing modelling studies of the ocean carbon cycle (eg: Menviel et al. 2012) but also what we might expect to find based on the proxy evidence that is referenced later in the Discussion (consider comparing to Yu et al. 2013). Considering the emphasis of the results on the change in the solubility pump, I would also like to see more written about expected changes in ocean circulation and what previous studies have found (eg. Born et al. 2010; Mokeddem et al. 2014). Trends in atmospheric CO2 are also discussed, but are not relevant to this study which compares two equilibrated climate states.**

Response: We thank referee#2 for providing new relevant references to our paper. We have substantially modified the introduction accordingly to referee#2 suggestions and added new and more relevant references.

Revision in the introduction section:

"Few model based studies examine the carbon cycle dynamics for the LIG period with a particular focus on the ability of models to simulate the transient changes in atmospheric CO2 concentration, which remains relatively stable around 270-280 ppm without displaying any trends (Lourantou et al.,

2010; Schneider et al., 2013). Most of these studies provide a better understanding of the land carbon budget  particularly highlighting the importance of temperature changes on the land vegetation and slow processes of CO2 change such as peatland carbon dynamics and CaCO3 shallow water accumulation (Schurgers et al., 2006; Kleinen et al., 2016; Brovkin et al., 2016). Although they are numerous studies that have analyzed the role of the ocean carbon cycle in regulating the atmospheric CO2, especially for the interglacial-glacial transition period (Ridgwell 2001; Sigman and Boyle 2000; Menviel et al., 2012), to the authors' knowledge, there is no study that investigate in details changes in marine carbon and nutrient cycling during the Eemian period of the LIG (125ka – 115ka).

With respect to changes in large-scale ocean circulation, reconstructions indicate that deep Atlantic circulation patterns and water mass geometries likely change over this interval, with a persistent mid-depth Atlantic ventilation of northern sourced waters (McManus et al., 2002; Mokeddem et al., 2014), while southern sourced waters expanded at depth toward 115ka (Govin et al., 2009). In addition to large-scale circulation changes, temperature-induced changes in carbon solubility pumps and biological production are expected to alter the ocean carbon budget, in particular in the interior ocean. Other changes such as sea-ice extent and ocean ventilation could also affect ocean carbon sequestration rate.

**P5L24 – This could be a little clearer. Perhaps clarify that DICdis and DICsat equate to preformed carbon, or the solubility pump. This should be noted before these components of the solubility pump are deconstructed and their calculation described.**

*Response: We agree and follow referee#2 suggestion by dividing Eq. (1) into two new equations at the beginning of the subsection 2.3*

Revision:

"In order to analyze the oceanic carbon cycle, dissolved inorganic carbon (DIC) is decomposed into its preformed and biological components (Eq. (1)), following Bernadello et al. (2014):

$$DIC^{tot} = DIC^{pre} + DIC^{bio} \tag{1}$$

The preformed component of DIC (DICpre) comprises satured and disequilibrium parts (Eq. (2)).

$$DIC^{pre} = DIC^{sat} + DIC^{dis}, \tag{2}"$$

**P6L11 - "We will first describe". Please be more explicit, change to something like "Section 3.1 will describe...". Again, for "The second section", please change to "Section 3.2" and also the sentence beginning with "Finally".**

*Response: We concur and have rephrased the paragraph in the section 3.*

Revision Sect. 3: "Section 3.1 describes the sea surface temperature and sea ice changes while changes in water mass properties are discussed in Sect. 3.2. The near surface changes that particularly influence the biological pump is addressed in Sect. 3.3 and global to regional oceanic DIC storage changes are

presented in Sect 3.4. The analysis is performed over the average of the last 50 years of the simulations. In addition, the global ocean is divided into three main basins: Atlantic, Indian and Pacific."

**P6L16 - "Simulated sea surface temperature (SST) during the 125ka experiment were warmer globally" with respect to what? Referencing Fig.1 would be appropriate here. I would also recommend a quick comparison to Hoffman et al. 2017 either here or in the discussion comparing the magnitude/ spatial pattern of SST change. This wouldn't require any new figures (particularly as the paper compares to PI SSTs) but simply a qualitative comparison.**

*Response: We have edited a completely new section on sea surface temperature, sea ice and mixed layer depth changes where we clarify the above statements.*

Revision in Sect. 3.1: "Our model simulates a global and annual increase of sea surface temperature (SST; +0.27°C) in 125ka relative to 115ka experiment."

*In addition we have also added the reference to Fig. 1 and a comparison to Hoffman et al., 2017 regarding our preindustrial run.*

Revision in Sect. 3.1: "We note that relative to the preindustrial, based on proxy reconstruction (Hoffman et al., 2017), our model simulates consistent spatial feature of annual SST anomalies at 125ka. It simulates (i) strongest warming in the high latitude, specifically in parts of the Southern Ocean, (ii) weak cooling in the low latitude, (iii) cooler SST in most of Indian Ocean, and (iv) warmer northeast Pacific, among others. Nevertheless, the amplitude of SST warming and cooling at specific sites tend to be weaker in our model. This feature appears to be common in other global models and could be attributed to their low spatial resolutions (Hoffman et al., 2017)."

**P6L16 - "but the changes varied spatially and seasonally, affecting both the ventilation and the nutrient supply at the surface". Please write in the present tense (change varied to vary). I would also like to see contours of Δmax mixed layer depth (or some other metric to diagnose changes in convection/ upwelling) overlaid on Figure 1, or perhaps Figure 2. It would be interesting to see if, with the retreat of Antarctic sea ice in 125ka, the location of deep and intermediate water formation is shifting with respect to 115ka. Based on discussion later in the paper of the Ferrari et al. 2014 sea ice mechanism, it might also be useful to see changes in sea ice extent on these figures and how this is affecting mixed layer/ convective depth and also the location of the formation of these water masses.**

*Response: The new Sect. 3.1 includes a new figure (Fig. 1) that shows the difference in SSTs between 125ka and 115ka in shade, the 50% of sea-ice extent during each season (DJF, MAM, JJA, SON) for both experiment (125ka as purple lines and 115ka as green lines) and the increase (decrease) of the mixed layer depth as thick black (blue) lines. More precisions have also been made on specific regional changes in SSTs.*

[Figure]

*Figure 1: Difference in Sea Surface Temperature (ΔSST) between 125ka and 115ka. Only significant differences (i.e.,with absolute value greater than the interannual standard deviation over the last 50 years in both 125ka and 115ka) are shown.The green and purple lines correspond to 50% sea ice extent in 115ka and 125ka, respectively. The black (blue) thick lines depict regions affected by a deepening (shallowing) of the mixed layer depth (with difference superior of 100m depth) in 125ka compared to 115ka.*

Revision in Sect. 3.1: "This warming is mainly simulated in the high latitudes (Fig. 1a-d) where higher SSTs are simulated throughout the year in the Southern Ocean, south of Greenland, the Norwegian sea and the northern part of the Pacific Ocean. This persistent warming in 125ka induces the sea ice to melt (Fig. 1, green and purple lines). This impacts directly the mixed layer depth (MLD) by leaving new areas free from sea-ice allowing for air-sea exchanges (Fig. 1, black lines in the Southern Ocean). In the Labrador Sea the mixed layer depth in 125ka is deeper of more than 100m than in 115ka. This is due to higher salinity simulated in this region in 125ka. At lower latitudes, the SSTs vary more spatially and seasonally. For example, in some parts of the Atlantic Ocean (North Atlantic drift, Equatorial and some sections of the Subantarctic near 45S band) cooler SSTs are simulated over several seasons (4SST < 0, Fig. 1a-b). While the North Atlantic drift depicts cooler SSTs during boreal winter and spring (Fig. 1a-b), colder SSTs last until the boreal summer in the Equatorial region of the Atlantic Ocean (Fig. 1a-c). The 45S latitude band remains cooler over the four seasons. Here, the MLD seems to be more controlled by changes in salinity instead of temperature distribution."

**P6L18 - "due to a weakening (strengthening) of the mixing process.". Be explicit – do the authors refer to a flattening of the isopycnals resulting in reduced turbulent mixing?**

*Response: In the revised paper, this sentence has been removed since SSTs where unable to explain changes in MLD. The latter are mainly influenced by salinity and sea-ice changes (see revision for the previous comment P6L16).*

**P6L19 - "In the Atlantic, cooler SSTs are simulated during boreal winter and spring ($\Delta$SST < 0, Fig. 1a-b), which allow for more upwelled nutrients to the surface (Fig. 2a) via an increasing of the mixed layer depth." - This is not true of the whole basin, be explicit where these changes occur. I would again re-iterate the usefulness of having changes in mixed layer depth/ convective depth contoured on these panels. The grammar of this sentence is also not quite right, please rephrase.**

*Response: In the revised paper, more details are provided on regional SSTs changes in the new Sect. 3.1. The link between SSTs and MLD has also been revised as mentioned in the response to the above comment (P6L18).*

**P6L20 - "This higher concentration of nutrients increases the biological production (Fig. 2b) under more favourable warmer blooming season in summer in 125ka ($\Delta$SST > 0, Fig. 1c)." I would note that the authors should refer to 'export production' explicitly, rather than just 'biological production' which may be misunderstood as net primary productivity.**

*Response: We have modified this wording in the updated version of this section (new Sect. 3.3) and refer explicitly to export production.*

Revision in Sect. 3.3: "Nevertheless, due to higher preformed and remineralized phosphate in the subantarctic water sinking region, more PO4 is advected through this 'ocean tunnel' to the equator and therefore leads to an increase in EPC (Fig. 5a, turquoise rectangle)."

**Furthermore, the region of the ocean that coincides with increased export production in Fig. 2B is actually cooler in 125ka (Fig. 1c).**

*Response: Thank you for pointing this out. After further analysis of the relationship between SST changes, MLD and export production, the changes in export production in the Equator are mainly controlled by changes occuring where these water masses are originated - the subantarctic water – instead of SST changes. We separated the section discussing changes in SSTs and carbon export and splitted it in two entities. The new Sect. 3.1 discusses the changes in SSTs, sea-ice and MLD, while the new Sect. 3.3 discusses the near surface productivity.*

**P6L23 – I find this sentence to be difficult to understand on first glance. Perhaps rephrase?**

Response: This sentence has been rephrased in the revised paper.

Revision in Sect. 3.3: "In the Atlantic Ocean, the Subantarctic water (45°S latitude band) corresponds in the model to the southern sourced intermediate water formation. This water mass sinks and reemerges along the Equator (Fig. 5, turquoise rectangles)"

**P6L26-27 – I would like to see further explanation regarding these 'ocean tunnels', particularly given that our current understanding of ocean dynamics in this region would suggest that intermediate water formed in the South Atlantic sector would take a very indirect route towards the equator, diminishing this signal (flowing north in the Malvinas current, until it is deflected eastward by the Brazil current, before some proportion is entrained into the Benguela Current and reaches the tropics). A reference or two would be sufficient here.**

*Response: Following referee#2 suggestion, we have added some more explanations and references to this "ocean tunnels". We also agree that the pathway between subantarctic intermediate water to the Equatorial region is complex, and potentially not well represented in the model. Nevertheless, detailed analysis of the ocean dynamic in this region is beyond the scope of our study.*

Revision in Sect. 3.3: "This pattern expected from models and modern observations shows that the intermediate and mode waters formed in the high southern latitudes feed the subtropical thermocline and act as a predominant source of nutrients important for sustaining low-latitude biological productivity (Gu and Philander, 1997; Sarmiento et al., 2004). We acknowledge that there is uncertainty in the complex pathways of the subantarctic water masses toward the Equator simulated in the model.

**I'm also curious if the authors investigated changes in equatorial/ low latitude winds, as the decrease in export production in the Eastern equatorial Pacific may also be in part due to a weaker Equatorial Undercurrent.**

*Response: Yes we have. In our simulations, the wind did not seem to have an important impact on the export production through the Equatorial Undercurrent. Even though in 125ka a slightly stronger wind is simulated in the northern part of the Equatorial Atlantic region and support the stronger export production in the equatorial Atlantic Ocean, in the southern part of this region, a weaker wind is simulated in 125ka compared to 115ka, despite having similar export production. In the Pacific Ocean, the changes in wind at low and equatorial latitudes does not show any clear pattern that could help explaining the changes in export production.*

Revision in Sect. 3.3: "Although there are changes in surface wind speeds in the Equatorial regions of Atlantic and Pacific basins, they do not explain the simulated changes in export production."

**P6L29 – Southern ocean upwelling is largely wind-driven and southern ocean density structure is controlled primarily by salinity rather than temperature. Changes in SSS are mentioned later on P7L30, but it is worth noting here that the change in the Southern Ocean's density structure is not purely due to an increase in SST, which is implied here.**

Response: The referee#2 is correct. In the new Sect. 3.1 the importance of SSS in the Southern Ocean are mentioned (see revision to comment P6L16).

**P7L5 - "Despite latitudinally homogeneous forcing..." Not sure what this means? The difference between the two climate states should only be the concentration of green-house gases and orbital forcing (which should affect high-latitude insolation). This difference in insolation is zonally homogeneous, but certainly differs with latitude.**

*Response: We thank the referee#2 for noticing this mistake. The forcings are indeed zonally homogeneous.*

Revision in Sect. 3.3: "Despite zonally homogeneous forcing we find a basinally heterogeneous response in both Subantarctic ventilation and in low latitude productivity which is similar to, albeit more extreme than, the basin specific response simulated for future warming and stratification (Moore et al., 2018)."

**P7L32 - 'As a net result, the water mass becomes younger in the SO because of a southward retraction of SSW and southward incursion of more and younger NSW.'. I think this isn't phrased quite correctly. The water mass becomes younger because of a retreat in Antarctic sea ice extent and influx of meltwater, (presumably) an increase in wind-driven upwelling/ mixing and an increase in bouyancy which in turn LEADS to a retraction of SSW.**

*Response: The referee#2 is quite correct, we rephrased the above statement in the revised paper. However changes in wind-driven upwelling are relatively small in the Atlantic section of the Southern Ocean due to relatively unchanged wind forcing between the two periods studied.*

Revision in Sect. 3.2: As a net result, the water mass becomes younger in the SO because of sea-ice retreat and the influx of meltwater, which in turn induces a southward retraction of SSW and southward incursion of more NSW.

**P8L3 - 'into the interior during the warmest period (Fig 3b).' - please refer to it explicitly as 125ka, or simply add it afterwards (ie: into the interior during the wamer period (125ka – Fig 3b))**

*Response: We follow the referee#2 suggestion.*

Revision in Sect. 3.2:While such large redistributions of northern and southern origin deep waters only

occurs in the Atlantic, these changes also influence water properties in the Indian Ocean due to the 'downstream' advection of younger deep water into the interior during the warmest period (125ka - Fig. 3b).

**P8L3 - 'northward in the Indian, the' - please add Ocean/ Basin.**

*Response: We agree and revised accordingly.*

**P8L4 - 'Indian deep water must also decrease (increase)' why is increase included in brackets here? There is no point of comparison.**

*Response: The increase included in brackets referred to the turnover rate previously mentioned between bracket. However, we agree that it can be misleading and therefore removed this comparison in the revised manuscript.*

Revision: "In the Atlantic Ocean, the Subantarctic water (40°S latitude band) corresponds in the model to the southern sourced intermediate water formation. This water mass sinks and reemerges along the Equator (Fig. 5, turquoise rectangles). This pattern expected from models and modern observations shows that the intermediate and mode waters formed in the high southern latitudes feed the subtropical thermocline and act as a predominant source of nutrients important for sustaining low-latitude biological productivity (Gu and Philander, 1997; Sarmiento et al., 2004b). We acknowledge that there is uncertainty in the complex pathways of the subantarctic water masses toward the Equator simulated in the model."

**P8L8 - 'the eastern side waters of the basin are simulated to be older in 125ka by as much as 300 years older.' remove 'older' at the end of the sentence.**

*Response: We agree and removed 'older', as suggested.*

**P8L8 - 'This older water masses are created in the Pacific SO and are predominantly affected by the strong increase in SST, increasing therefore the stratification' - 'This' should be 'These' and 'increasing therefore the stratification' makes no grammatical sense. Please rephrase.**

*Response: We have rephrased the above statement.*

Revision in Sect. 3.2: "While the western side is also influenced by the younger water masses formed in the Atlantic Ocean, the eastern side waters of the basin are simulated to be older in 125ka by as much as 300 years. These older water masses are formed in the Pacific SO and may be affected by a flattening of the isopycnals south of 60°S in 125ka (Fig. 4d, gray lines). This flattening of the isopycnals is influenced by both sea-ice melting and higher SSTs, and suggests a stronger stratification and weaker subduction rate"

**P8L9 - 'In the northern hemisphere the younger waters are due to to cooler SST (Fig. 1).' I'm sceptical that a slight cooling in the surface sub-tropical and equatorial Pacific would produce a sufficient increase in circulation to decrease the water mass age of the North Pacific basin down past a depth of ~1000m. I suggest investigating the meridional overturning streamfunction for the Pacific, but more importantly, identifying a mechanism to explain why the deep North Pacific is so much younger. I suspect that, similar to the Southern Ocean, a reorganisation of water masses has simply resulted in this region of the deep ocean to be better ventilated, and has very little to do with SST in the North Pacific.**

*Response: In the revision process, we have investigated further the changes occuring in the Pacific Ocean. It seems that the modification of the upper cell of the overturning circulation (stronger in 125ka than 115ka) can give a potential explanation for the reduction of DIC at mid-depth.*

Revision in Sect. 3.2: "Finally, the Pacific intermediate waters are simulated as younger in 125ka when compared to 115ka (Fig. 3c), which is consistent with the strengthening of upper cell of the overturning circulation previously mentioned for this region."

[Figure]

*Figure R1: Overturning Stream function of the Pacific + Indian Oceans (in Sv). Negative values (blue colors) indicates an anti-clockwise circulation. The upper cell overtuning circulation is stronger in 125ka, suggesting better ventilated mid-depth water.*

**P8L16 - 'They are significantly different and basin specific responses of deep water ventilation rates and water mass distribution to uniform changes in forcing.' This sentence is not at all clear (what does 'they' refer to?), please rephrase.**

*Response: We concur and have modified the paragraph in Sect. 3.2.*

Revision in Sect. 3.2: "In response to the different forcings between 125ka and 115ka, significant changes are simulated in deep water ventilation rates and water mass distribution in the three basins. While the responses in the Atlantic and Indian Oceans have some similarity (better deep water ventilation), the Atlantic basin seems to be the most sensitive and simulates also water mass geometry changes. However, the ventilation rate in the Pacific Ocean depicts an opposite sign of change than the Atlantic and Indian Oceans. In the next section we discuss how these modification impact the near surface productivity in our model."

**P8L32 - 'The ΔDIC tot of the Indian basin resembles that of the Atlantic at depth, where the soft-tissue and disequilibrium components simulate the strongest decrease (Fig. 5c, green and purple bars). However, the saturation component depicts persistent negative ΔDIC below 1000 m depth, thereby accounting for the second most important component of the decrease throughout the water column.' These two sentences could lead to confusion, as the authors initially state that the Indian Ocean resembles the Atlantic where the soft-tissue pump and disequilibrium components are responsible for the greatest decrease in DIC, however the very next sentence correctly identifies that the saturation component is in fact the second most important component. Please rephrase to enhance clarity.**

*Response: We have considered the referee#2 suggestion and removed the first sentence.*

Revision in Sect. 3.4: "The $\Delta DIC^{tot}$ of the Indian basin also depicts a strong reduction in 125ka relative to 115ka. Here the soft-tissue and saturation components simulate the strongest decrease (Fig. 6c, green and blue bars)."

**P9L9 - 'zonally averaged' I'm curious if there is a reason why these plots do not show zonally integrated values of DIC? While I'm sure this would not change the interpretation of the results, it is, in my opinion, the more appropriate depiction of ΔDIC. This is particularly true when each panel should be representing the components of the sum total change in panel a) of each figure.**

Response: Due to the irregularity of the spatial model grid (vertically and horizontally) the zonally integrated values of DIC would be biased with higher (lower) mass in thickest (thinnest) layers. Even if we interpolate to regular grid, the zonal extend of each latitude volume also varies. For example in the Pacific, the Equatorial latitude exhibits larger volume and hence mass than in the North Pacific. The new Fig. 6 (previous Fig. 5), shows the integrated values of DIC vertical profile. The purpose of Figs.

7, 8 and 9 is to illustrate the meridional pattern of change in DIC and relate it with different water masses, which we we think is shown better in concentration.

**P9L13 - 'At depth, the changes in SSW and NSW lead to a decrease in younger water masses, hence less remineralized organic matter.' I find this confusing – shouldn't older deep water masses contain more remineralised DIC due to their decreased ventilation rate?**

*Response: True. We have corrected the sentence in the revised manuscript in Sect. 3.4.*

Revision in Sect. 3.4: "At depth, the slightly deeper AMOC in 125ka (compared to 115ka), leads to better ventilated mid-depth to bottom waters in the Northern Hemisphere leading to less remineralized organic matter by reducing the water mass age."

**P9L15 - 'Positive change in $\Delta$DIC tot also arises' where?**

*Response: We have updated the sentence and added more precision in the revised manuscript.*

Revision in Sect. 3.4: Positive changes in $\Delta DIC_{tot}$ in near surface waters and bottom water at 20°N also arise from the soft-tissue and carbonate signal due to the increase of the alkalinity (not shown here) and slightly older water masses along the African coast.

**P9L18 - 'The latter' It is unclear what is being referred to here.**

*Response: We have modified the sentence in Sect. 3.4.*

Revision in Sect. 3.4: "This change in disequilibrium is due to "[...].

**P9L19 - 'The NSW water mass, formed in the North Atlantic, is generally more subject to biological production during its near surface northward transport before sinking into the interior than the SSW' please provide references to support this claim.**

*Response: A reference has been provided.*

Revision in Sect. 3.4: "The NSW water mass, formed in the North Atlantic, is generally more subject to biological production during its near surface northward transport before sinking into the interior than the SSW (Duteil et al., 2012)."

**P9L25 - 'due to weaker ventilation induced by stronger SSTs in the Labrador and Nordic sea.' please reference Fig. 1 here. Also, 'stronger' is an inappropriate word here, consider replacing with something like 'warmer'.**

*Response: This statement was meant to explain the changes in DIC$^{dis}$ in the upper layers of the Atlantic basin. We thank the referee for pointing out this statement and have made some adjustment since (after careful look) the SSTs changes couldn't explain the changes in ventilation. Instead, it could be link to the intrusion of more SSW (less disequilibrated than NSW) at the near surface as indicated in Fig. 4C at 800m depth.*

Revision in Sect. 3.4: "However, the upper layers of the Atlantic Ocean are mostly simulated with higher DIC$^{dis}$ (positive DIC$^{dis}$; or less disequilibrium in 125ka). This could be induced by more SSW (less disequilibrated than NSW) entering further north the Atlantic basin at the near surface as suggests Fig. 4c."

**P9L28 – Indian *Ocean.**

Response: We agree and revised as suggested.

**P9L35 - 'Changes in the bottom water DIC soft can be attributed primarily related to the difference in the disequilibrium effect due to stronger carbon export in the Southern Ocean (Fig. 2b) and to a slight decrease in the saturation component. ' Remove 'related'.**

*Response: We concur and also corrected a typo from DIC$^{soft}$ to DIC$^{tot}$.*

Revision: "Changes in the bottom water DIC$^{tot}$ can be attributed primarily to the difference in the disequilibrium effect which is probably affected by larger influence of NSW in 125ka relative to 115ka as described for the Atlantic basin, and to a slight reduction in the saturation component. In addition, the disequilibrium component might also be influenced by stronger carbon export in the Southern Ocean (as the positive DIC$^{soft}$ and Fig. 5a suggest)."

**P10L10 - 'This lower organic remineralization arises from an increase of the ventilation around 30N and a potential increase of the upwelling generating younger water masses (Fig. 3c)' This sentence is misleading – there isn't increased ventilation at 30N, rather, water masses at 30N are better ventilated (likely via some pathway that results in upwelling in the Southern Ocean). Please rephrase to avoid confusion.**

*Response: After careful consideration of the referee#2 comment, we agree that the SSTs would only have a small impact on deep ventilation. Instead, we relate this changes in deep ventilation to the increase in the overturning circulation.*

Revision: "This decrease in biogenic carbon is induced by better ventilated water mass around 30°N (Fig. 3c), which may come from the increase of the overturning circulation in the upper cell of the Pacific Ocean."

**Also, if I understand this correctly, ' This lower organic remineralization' refers to the decrease in DICsoft rather than an actual decrease in the remineralization rate in the ocean. If this is the case, consider rephrasing this to something along the lines of 'this decrease in biogenic carbon ...'.**

*Response: The referee#2 is correct. The statement has been reformulated (see previous comment).*

**P10L11 - 'This is in good agreement with the increased carbon export production (Fig. 2b)' But export production is largely unchanged or negative in the Pacific Ocean according to Fig. 2b.', inducing positive ΔDIC tot near the surface, and with the cooler SST depicted in Fig. 1' Negative SSTs are only evident in Boreal winter and spring – does this outweigh the effect of considerably warmer SSTs in summer and autumn?**

*Response: The changes in $DIC^{dis}$ in the Pacific Ocean are still unclear as it can be affected by many different factors. However, in the revised manuscript we have suggested few processes that might contribute to this changes.*

Revision: "However, the bottom waters seems to be the most affected and become more undersaturated in 125ka relative to 115ka. This may be influenced by the slowing down of the subduction process in the Southern Ocean, induced by the flattening of the isopycnals. The possible higher carbon export production in the SO south of 60°S (Fig. 5a) may also pushes the water mass further out of equilibrium."

**P10L21 - 'This SSW mainly composes the Pacific interior ocean.' please rephrase.**

*Response: This statement has been removed as it was not needed in the new version of the paragraph (see above response).*

**P10L30 - 'Significant decreases of the ocean carbon' replace 'of the' with 'in'.**

*Response: Done.*

**P10L31 – This is a good summary, but I would suggest that instead of saying 'Most of this decrease', give the exact %. I would also like to see an extra sentence at the end of this summary paragraph highlighting the mechanisms responsible for the change in ocean ventilation. This could include AMOC strength, changes in SSW/NSW, changes in AABW/IW formation, sea ice changes etc.**

*Response: We follow the referee#2 suggestion and have added the following statement.*

Revision in Sect. 4: "More than 48% of this decrease is induced by the reduction of the biological pump. This decrease is found to be mainly driven by the shorter residence time of interior deep water

masses arising from changes in Southern Ocean sea-ice extent that influence the NSW/SSW water mass geometry, in addition to changes in overturning circulation in the Atlantic (deeper and slightly stronger) and Pacific basins (stronger upper cell and weaker bottom cell)."

**Comparison to proxies regarding water masses is good, though I would suggest one or two extra studies that could be referenced during the discussion (please see list at the end of this document).**

*Response: We concur and have added new references to the revised text. However, following other referees comment and to keep the focus of our study, we removed previous Fig. 9 and only refer to the literature instead of describing precisely d13C changes.*

Revision: "Similar to our results, expanded SSW in the late compared to early LIG is inferred from such reconstructions to explain bottom water $\delta^{13}C$ decreases in different regions proximal to the Southern Ocean (Govin et al., 2009; Ninnemann et al., 2002). Bottom water $\delta^{13}C$ reconstructions indicate less influence of NSW in the deep South Atlantic at 115ka than at 125ka (Ninnemann et al., 1999; Govin et al., 2009). In addition, persistent (millennial-scale) mid-depth NSW ventilation extending from the LIG and into the subsequent glacial inception is also suggested based on proxy reconstructions (McManus et al., 2002; Mokeddem et al., 2014) and is consistent with model simulations (Born et al., 2011; Wang et al., 2002). In our study, even though the AMOC is simulated slightly stronger and deeper in 125ka, vigorous AMOC persists during 115ka ventilating the North Atlantic mid-depth. We also note that our model may not properly represent North Atlantic overflows due to its sparse resolution. This can further add uncertainties to North Atlantic water ventilation."

**P11L24 - 'A sense of the scale of the changes model simulations can be gained through comparison to previous modelling efforts where atmospheric CO 2 was not fixed.' this sentence does not make sense, please rephrase. I'd also like to see comparisons to Menviel et al. 2012 here.**

*Response: We concur and modified that sentence.*

Revision: "Our estimated changes in ocean carbon budget is in the range of previous modeling effort studies that also suggest weaker ocean carbon storage during the beginning of the LIG (125ka) relative to the glacial inception (115ka)."

**P12L1 - 'obtained a difference in atmospheric CO 2 concentration and terrestrial carbon storage of about 40 PgC and 350 PgC, respectively' If you refer to an atmospheric concentration, then please change the 40 PgC to ppm equivalent, or alternatively, rephrase to refer to "atmospheric and terrestrial carbon storage" instead. The authors should also be clearer here: is this +350Pg and -40Pg? Similarly, the following line claims that a change in ocean carbon of 310 Pg corresponds well to your findings of -314.1 PgC. These values are opposite, is the former supposed to be -310?**

*Response: We modified the statement following the referee#2 suggestion.*

Revision: "Another study by Schurgers et al. (2006) obtains a difference in atmospheric and terrestrial carbon storage of about 40 PgC and 350 PgC, respectively, between the onset and end of the LIG. This potentially translates to a weaker ocean carbon storage of 310 PgC at the onset compared to the late LIG, which corresponds well in absolute magnitude with our findings of 314.1 PgC decrease in 125ka."

**P12L4 - 'However, the changes in CO 2 concentration in the atmosphere that they simulated steadily increases' This is poorly worded and gives impression that the rate of change of atmospheric CO2 is varying somehow. Please rephrase. Something along the lines of "However, their simulated atmospheric CO2 steadily increases over this period".**

*Response: We concur and revised the sentence as suggested.*

**P12L31 - ' The ongoing global warming raises questions about the oceanic carbon sink and its efficiency under a warmer climate condition.' Be specific. 'Ongoing anthropogenic warming...' and remove 'The' from the start of the sentence.**

*Response: We concur.*

**P12L32 - 'In this study, we use a fully-coupled NorESM' please change 'a' to 'the' (unless of course there is more than one version of Nor-ESM, which should then bestated).**

*Response: We concur have revised the sentence.*

Revision: "we use the fully-coupled NorESM model"

**P13L1 - 'We focus on the differences that occurred in 125ka in comparison to 115ka, specifically the differences at global and basin scales.' This is vague – be explicit that you're looking at changes in the ocean carbon cycle.**

*Response: We concur.*

Revision: "We focus on the differences in ocean carbon cycle that occur in 125ka in comparison to 115ka, specifically the differences at global and basin scales."

**P13L2 - 'Thereby, to our knowledge, it is the first attempt in elucidating the biogeochemical and physical processes that are responsible for the ocean carbon inventory changes under warmer climate conditions during the LIG.' Please see Menviel et al. 2012.**

*Response: This is true. Menviel et al., 2012 did a great work in describing interglacial-glacial ocean*

*carbon cycle. However, the window allocated to period studied here (125ka – 115ka) is small compared to their timescale (125ka to 0ka), which makes it difficult to assess changes in detail as well as regional for that period. However, we revised the above statement accordingly.*

Revision: "We provide a detailed description of the biogeochemical and physical processes that are responsible for the ocean carbon inventory changes under warmer climate conditions during the LIG at the temporal and spatial scales discussed here."

**P13L24 - 'The western Pacific is influenced by water masses coming from the Pacific Southern Ocean, with a warmer SST that hinders the ventilation and increases the residence time of the interior water masses on the eastern side of the basin.' On P8L6, the authors state that the Western Pacific is ventilated by water originating from the Atlantic. Either this section or that in section 3.2 needs to be clarified/ corrected.**

*Response: We thank the referee#2 for noticing this error. We have corrected it in the revised manuscript.*

Revision: "The eastern Pacific Ocean is influenced by water masses coming from the Pacific sector of the Southern Ocean"

Minor typographical errors/ suggestions

**Please capitalise names of basins/ hemispheres throughout the manuscript.**

*Response: Done.*

**P1L7 - "We find a considerable weaker ocean dissolved..." this sentence does not make grammatical sense. I would suggest changing it to "We find considerably reduced ocean dissolved..."**

*Response: Done.*

**L14 - "...restricting the extend of DIC rich southern sourced water reducing the storage of biological remineralized carbon at depth." Typo. 'extend' should be 'extent'. Furthermore, consider rephrasing as follows "...restricting the extent of DIC rich southern sourced water, thereby reducing the storage of biological remineralized carbon at depth.**

*Response: We have revised the text as suggested.*

**P1L20 - "If the anthropogenic greenhouse..." remove 'the'.**

*Response: Done.*

**P1L23 – consider rephrasing "For this, the changes in the the warm Eemian period may be considered as an analog for a future warmer climate." to "The changes in the the warm Eemian period may therefore be considered an analog for a future warmer climate.**

*Response: We concur.*

**P3L26 – prescribed should be prescribes.**

*Response: We concur.*

**P3L27 - "(>100m depth)" should this not be <100m depth? Or are you referring to the sinking rate of the particles that have left the euphotic zone?**

*Response: We modified the statement to make it clearer.*

Revision: "(above 100 m depth)"

**P3L31 - 'vertical advection': convection?**

Response: We have revised this statement.

Revision:".... vertical mixing and advection within the sediment layers."

**P4L24 - 'the dissolved inorganic carbon'. 'the' is not necessary.**

*Response: We remived "the".*

**P5L25 – Remove 'mostly' - vague language.**

*Response: Done.*

**P6L13 - 'Each analysis have' - please rephrase.**

*Response: We have rephrased.*

Revision: "The analysis is performed"

**P7L2 - 'southern hemisphere' should be Southern Hemisphere.**

*Response: We concur.*

**P7L17 - 'translates to a stronger ventilation rate age and' remove 'age'.**

*Response: Done.*

**P7L22 - 'since the SST in the SO is rather warmer in 125ka' remove 'rather'.**

*Response: We have added a new subsection that describes SST changes in the revised manuscript (Sect. 3.1). This sentence has been removed.*

**P7L26 - 'byFerrari et al. (2014)' space missing**

*Response: Space added.*

**P7L26 - 'study,Luo et al...' space missing.**

*Response: Space added.*

**P8L27 - 'main contributors for the weaker carbon inventory' replace weaker with reduced.**

*Response: We concur.*

**P9L3 - 'simulating a strong positive difference of about +18 PgC' strong with respect to what? I would suggest removing such descriptive words.**

*Response: We concur and removed "strong".*

**P9L6 - 'However, the saturation component has also a considerable influence' 'has also' should be 'also has'.**

*Response: We concur.*

**P9L29 - 'Only in the region that may correspond to the AAIW the simulated 4DIC tot are positive (Fig. 7a, red shade).' Poor grammar, please rephrase.**

*Response: We have rephrased.*

Revision: "Positive ΔDICtot are nevertheless simulated in the region that may correspond to the AAIW (Fig. 8a, red shade)"

**P11L6 - 'has been previously been inferred' remove second 'been'.**

*Response: We concur.*

**P11L27 - 'but at slightly weaker decrease than in our study.' grammatically incorrect. Please rephrase.**

*Response: We have rephrased.*

Revision: "but at slightly weaker amplitude than in our study."

**P12L8 - 'our model study shows an heterogeneous response' 'a' not 'an'.**

*Response: We concur.*

**P12L10 - 'Such heterogeneous response of the biogeochemical divide have also been highlighted by Moore et al. (2018) for future projections under warmer climatic conditions.' Consider rephrasing to 'Such a response...' and '...has also been highlighted...'**

*Response: We concur.*

Revision: "Moore et al. (2018) also highlight such biogeochemical divide response for future projections under warmer climatic conditions.

**P12L11 - 'This implies that changes in the biogeochemical divide could somewhat be similarly impacted from past and future anthropogenic CO 2 forcings' Poorly phrased, please correct.**

*Response: We have rephrased.*

Revision: "This implies that future anthropogenic CO2 forcings may have a similar impact on the biogeochemical divide with that of past forcings."

**P12L12 - "Reconstruct and understand the pattern and signs of past responses of large scale**

**productivity to climate forcing are therefore critical for assessing not only the sign but also the sensitivity of different regions to climate change." Poorly phrased, please re-write.**

*Response: We have rephrased.*

Revision: "Therefore, reconstructing and understanding the large scale productivity responses to past climate forcing are critical for assessing both global and regional sensitivity of the ocean carbon dynamic to climate change."

**P12L15 - 'Factors that could influence ocean carbon storage including sea level, riverine input of nutrients, and atmospheric dust loading, which are all set to preindustrial levels in our simulations, but may have been different in the LIG.' remove 'which', otherwise sentence does not make grammatical sense.**

*Response: We concur and removed "which".*

**P12L23 - 'Our quasi-equilibrated model simulations for 115ka and 125ka, also lack ice sheet and the corresponding freshwater input variability, do not address such shorterterm changes that could affect the ocean carbon inventory (Stocker and Schmittner, 1997).' *and do not address.**

*Response: We concur and added "and".*

**P13L32 - 'a priori' typically written in italics, though this is only a suggestion.**

*Response: We concur and used italics.*

**Table 1: 'southern sources water' correct to 'sourced' I'm also curious as to why values for NSW were not also included in this table and discussed in the manuscript.**

*Response: We concur and corrected as suggested. In addition, we decided to mainly focus on the SSW since this water mass is strongly affecting the ocean interior in all three basins, while NSW have relatively low influence on the interior Indian and Pacific Oceans.*

**Figure 2. The authors refer to this box as a 'turquoise rectangle' here but as a green rectangle elsewhere in the manuscript. Please be consistent and change one or the other.**

*Response: This has been corrected.*

**Figure 4. Please include units for the neutral density contours in the caption.**

*Response: The unit has been added.*

**Figure 5. Again, I question the utility of averaging over 500m intervals rather than summing. Particularly when dealing with bulk differences. Is there a reason for this?**

*Response: The purpose is to show vertical profile in DIC difference and illustrate in which interior depth the largest change is simulated. Previously Figure 5 actually does not show "average", but rater the sum of all masses within the 500m intervals (hence PgC units). The sum of difference for all integrated depth are also shown in the top-left corner of each panel.*

---

## Author Response (AR1)

We thank the editor L. Menviel for the time she spent processing our manuscript. Below is the response to her comments.

**In addition, all reviewers raised the issue of the processes leading to the significant loss of carbon at 125 ka, and I feel these processes need to be appropriately highlighted in your revised version.**

*Response: We have added few statement accordingly.*

Revision in abstract: "The biological pump is mainly driven by changes in interior ocean ventilation timescales, but the processes controlling the changes in ocean DIC disequilibrium remain difficult to assess and seem more regionally affected. While the Atlantic bottom water disequilibrium is affected by the sea-ice induced SSW/NSW organization, the upper layer changes remain unexplained. Due to its large size, the Pacific accounts for the largest DIC-loss, approximately 57% of the global decrease. This is largely associated with better ventilation of the interior Pacific water mass."

**Enhanced Southern Ocean ventilation (is AABW stronger in your simulation?) as is seen in Fig.3 and in Fig.2 of AC3 would lead to a carbon loss from the deep ocean (and particularly the Pacific) (see for example Menviel et al. 2014 (Paleo), or Menviel et al., 2015 (gbc)).**

*Response: Our simulations show clearly that under warmer climate (125ka) the interior ocean is better ventilated than under colder climate state (115ka). The mechanisms leading to this changes in ventilation rate seems however complex and our model outputs does not allow us to identify it as purely coming from the changes in AABW, but rather in changes of water pathways. Nevertheless, reduction in ventilation we simulate are consistent with previous studies (Menviel et al. 2014, 2015).*

Revision in Sect. 5: "We find that the global ocean carbon budget decreases during the warm (125ka) period by 314.1 PgC and are related mainly to better ventilation in the interior ocean. The Pacific Ocean has the largest reduction and accounts for 57% of the global DIC loss. The response of the Pacific ventilation in a warmer climate shown in this study is consistent with previous studies (Menviel et al., 2014, 2015)."

**This loss is probably enhanced by the constant atmospheric CO2 forcing and the higher SST.**

*Response: It is true. While higher atmospheric $CO_2$ concentration lead to an oceanic uptake, higher SSTs pushes the Ocean toward outgassing. Both of these processes are included in the computation of $DIC^{sat}$. However in our study, this component seems to be more sensitive to changes in preformed ALK as mentioned in Sect. 2.3. Therefore, higher preformed ALK lead to higher DIC and vice versa.*

References:

[revised manuscript text omitted]

---

## Author Response (AR2)

We thank very much editor L. Menviel for her additional comments that helped improve our manuscript. Below is the answer to L. Menviel latest comments.

**Throughout the manuscript: Please replace all "in 125ka" by "at 125ka"**
*Response: Changes have been made in the revised manuscript accordingly.*

**P1, last line: "due to changes"**

Response: Done.

**P2, L.14: "with a persistent mid-depth Atlantic ventilation of northern sourced waters". Could you please reformulate this? Do you mean that even if NADW transport varied across the LIG, there was always sustained NADW formation?**

Response: Yes, this is what the model simulates. We have reformulated this statement accordingly.

Revision: "With respect to changes in large-scale ocean circulation, reconstructions indicate that deep Atlantic circulation patterns and water mass geometries likely change over this interval. While mid-depth ventilation of northern sourced waters is maintained in the North Atlantic by the sustained NADW formation throughout the LIG (McManus et al., 2002; Mokeddem et al., 2014), southern sourced waters expanded at depth toward 115ka (Govin et al., 2009)."

**P4, L. 17: "two experiments"**

*Response: Done.*

**P4, L. 17: "by about 32 PgC"**

Response: Done.

**P6, L.16-17: "This impacts directly the mixed layer depth (MLD) by leaving new areas free from sea-ice allowing for air-sea exchanges". Please reformulate that sentence as it does seem grammatically correct, and take out the "s" at the end of "exchange"**

*Response: We have reformulated the statement accordingly.*

Revision: "Consequently, in the areas where sea-ice extent is reduced, there is more air-sea gas exchange and increase in mixed layer depth (MLD; Fig. 1, black lines in the Southern Ocean)"

**P7, L.11: "These changes"**

*Response: Done.*

**P7, L.22: "compared to"**

*Response: Done.*

**P7, L.24-26: I am not convinced by this sentence: the mixed layer depth is greater in austral winter in the Atlantic sector of the Southern Ocean at 125ka than at 115k. I would guess this leads to increased deep ocean convection, which would explain the lower ages in the Atlantic.**

*Response: That is true. We have modified this statement accordingly.*

Revision: "As a result of sea-ice retreat, the winter mixed layer depth in this area increases, inducing younger water mass age in the Southern Ocean."

**P7, l.28: "occur"**

*Response: Done.*

**P8, L. 4-5: It is unclear to me what you really mean here. I don't really see a poleward shift of SSW (which should form close to the Antarctic shelf) and I don't understand what you want to say about the water coming from the subtropical gyre.**

*Response: The polward shifting is small (approximately 2 degrees southward). A slight southward extension of the subtropical gyre could bring more water from the tropics to the Pacific section of the Southern Ocean. The concentration of phosphate of these tropical surface water masses is lower than that of the SSW-origin. Hence, slightly more tropical waters entering the SO could also decrease the phosphate availability in this region. We have modified the statement to make it clear.*

Revision: "In addition, the SSW boundary seems to be shifted southward from 58°S at 115ka to 60°S at 125ka (Fig. 3b,d, light purple dashed-lines). This may suggest that more surface water originating from the subtropical gyre (depleted in phosphate) enters the Pacific section of the Southern Ocean."

**P8, L.30 to p9, L.3: Shouldn't this be supported by showing ocean currents and their changes?**

*Response: From the model outputs it is not possible to diagnose the exact pathway of this complex circulation depicting intermediate waters formed in the Southern Ocean and upwell in the Equatorial regions. In order to do this, a separate lagrangian model tracer is needed. However, we show that the changes in circulation and wind speed are not consistent with the simulated changes in export production. On the other hand, changes in preformed nutrient appear to be consistent with the changes in export production, therefore we consider this mechanism as one potential hypothesis that could explain the changes in export production. We have revised the statement in the paper.*

Revision: "Although there are changes in surface wind speeds (not shown here) in the Equatorial regions, they do not explain the simulated changes in export production (i.e., stronger wind speed in the Equatorial Pacific Ocean and weaker wind speed in the Equatorial Atlantic region at 125ka). However, nutrient changes in the Atlantic and Pacific sections of the Southern Ocean are consistent with the changes in export production and therefore we suggest the following mechanism. In the Atlantic Ocean, the Subantarctic water (45°S latitude band) corresponds to the southern sourced intermediate water formation in the model. This water mass sinks and reemerges along the Equator (Fig. 5, turquoise rectangles). This pattern expected from models and modern observations shows that the intermediate and mode waters formed in the high southern latitudes feed the subtropical thermocline and act as a predominant source of nutrients important for sustaining low-latitude biological productivity (Gu and Philander, 1997; Sarmiento et al., 2004b). We acknowledge that there is uncertainty in the complex pathways of the subantarctic water masses toward the Equator simulated in the model."

**P9, L.5-6: Please remove: "where subantarctic waters upwell again."**

*Response: Done.*

**P9, L.7-8: I am really not sure this has been shown here. There is not enough information in the manuscript to clearly show that changes in productivity in each basin is due to changes in the ventilation of the Southern Ocean thermocline and not to changes in the strength of the upwelling in these areas (no information is given on winds or ocean currents).**

*Response: Thank you for pointing this out. Following your suggestion, we have also looked closer to the simulated surface wind speed and the upwelling strength of the equatorial regions with the simulated production changes. Changes in vertical mass flux and wind speed do not support the changes in export production. Figure R1 shows that the position of the upwellings are similar at 115ka and 125ka. More importantly, the regions that shows strong increase (decrease) in EPC in the Atlantic (Pacific) Ocean at 125ka compared to 115ka are not consistent with the changes in vertical flux depicted in Fig. R1c. For instance in the Equatorial Atlantic region, a weakening of the vertical mass flux is simulated, while the export production increases. Similarly in the Equatorial Pacific region where lower export productions are simulated at 125ka (south and north of the equator line), the vertical mass flux depict is relatively increasing while the export production decreases. In the same way, changes in wind speed (Fig. R2) in the Equatorial Atlantic region show a relative decrease in 125ka compare to 115ka, while it seems to be slightly increased in the Equatorial Pacific Ocean. We have added few statements in Sect. 3.3 with more explanation (see revision of previous comment P8, L.30 to p9, L.3).*

[Figure]

*Figure R1: Vertical volume flux (positive depicts upward) at (a) 115ka, (b) 125ka and (c) 125ka-115ka*

[Figure]

*Figure R2: Difference in wind speed at 10m above the ocean surface between 125ka and 115ka. NaN values are displayed when the difference is smaller than the standard deviation.*

**P11, L.6: "At 115ka"**

*Response: Done.*

**P12, L.10: "persists until 115ka"**

*Response: Done.*

**P12, L. 28: Please note that, contrarily to your study, in Brovkin et al., (2016), the ocean carbon content decreases between 125 and 115 ka due to changes in shallow water carbonate precipitation (a process not included in your model, correct?).**

*Response: We thank the editor pointing this out. Yes that is correct. This process is not included in our model. We have modified this section accordingly.*

Revision: "On the contrary, Brovkin et al. (2016) simulate more ocean $DIC^{tot}$ at 126ka than 115ka using simpler EMIC models. This increase is mainly attributed to the shallow water carbonate precipitation implemented in their model. This process is not included in our model and could explain the differences in the results. However, their simulated change in atmospheric $CO_2$ after 121ka is in the opposite direction (increasing of roughly 20 ppm) of the atmospheric trend (a decrease of roughly 3 ppm) observed in the ice core data. This suggests that either (or both) the terrestrial or the oceanic carbon reservoir does not take enough carbon toward the end of the LIG in order to bring down the atmospheric $CO_2$."

**P12, L.30-33: I think that these 2 sentences are not correct. Please note that in the simulations presented by Brovkin et al., 2016 , there is a global increase in land carbon due to peat accumulation (~32 PgC), while the vegetation loses carbon (~13 PgC). As atmospheric CO2 only changes by a few ppm between 125 and 115ka, the increase in ocean carbon content simulated in your study should be compensated by ~300PgC loss of terrestrial carbon. It is thus hard to conclude that changes in ocean carbon are driving the atmospheric CO2 concentration.**

*Response: These two sentences have been removed and this paragraph has been adjusted as presented in the above revision.*

**P15, L.8-9: There is something wrong with that sentence, please reformulate.**

*Response: The statement has been reformulated.*

[revised manuscript text omitted]

---

## Author Response (AR3)

We thank very much editor L. Menviel for her additional comments that helped improve our manuscript. Below is the answer to L. Menviel latest comments.

**1) Regarding EPC changes and their link to the upwelling. Thanks for providing maps of vertical transports and winds. Indeed the annual wind changes might be small, but it seems that the changes in vertical transport could impact EPC in the EEP and EEA… It is difficult for me to judge with what is available, but could you please check again.**

Response: In order to make it clearer we have updated the previous Fig. R1c and have added the changes in carbon export as contour lines (Fig. R3). In the Pacific Ocean the regions affected by a decrease in carbon export production (blue lines) depict an increase in vertical mass flux (red shade), which is the opposite process expected. Therefore, we suggest that this decrease in carbon export production could potentially come from a signal in the Southern Ocean where these water masses originate. Similarly, in the Atlantic basin, the regions depicting an increase in carbon export production (black lines) show a relative decrease in vertical mass flux. Only in the 8°S-16°S area, the increase in vertical mass flux may also have a positive influence on the carbon export production.

[Figure]

**Figure R3:** *Difference in Vertical volume flux (positive depicts upward) between 125ka and 115ka. The blue (black) lines depict a decrease (increase) by -4 g C m⁻² yr⁻¹.*

**Please check your statement about changes in wind speed in Section 3.3. p8, as this does not seem fully consistent with the figures you provided.**

*Response: We have revised the statement in Section 3.3 accordingly to Fig. R3.*

Revision: "Although there are changes in surface wind speeds and vertical mass fluxes (not shown here) in the Equatorial regions, they do not consistently explain the simulated changes in export production (i.e., stronger wind and upwelling in eastern Equatorial Pacific at 125ka compared to 115ka, where a decrease in export production is simulated; in parts of the Equatorial Atlantic some increase in export production may be related to the simulated stronger upwelling at 125ka)."

**2) In section 3.1., in the sentence you modified (L.4-5, p6): please add "an" before "increase": "..and an increase in mixed layer depth…"**

Response: Done.

**3) P8, L. 9: Since 125ka was before 115ka I would suggest to rephrase as:**

**"SSW boundary seems to be slightly poleward at 125ka (~60S) compared to 115ka (~58S)."**

*Response: We have modified the sentence accordingly.*

Revision: "In addition, the SSW boundary (Fig. 3b,d, light purple dashed-lines) seems to be slightly poleward at 125ka (~60°S) compared to 115ka (58°S)."

**4) P13, Please modify the last sentence of the amended section as follows: "…the LIG in order to simulate a decrease in atmospheric CO2 content"**

*Response: Done.*

[revised manuscript text omitted]